# PAC-Bayes Bounds for Multivariate Linear Regression and Linear Autoencoders

## Abstract

Linear Autoencoders (LAEs) have shown strong performance in state-of-the-art recommender systems. Some LAE models, like EASE, can be viewed as multivariate (multiple-output) linear regression models with a zero-diagonal constraint. However, these impressive results are mainly based on experiments, with little theoretical support. This paper investigates the generalizability – a theoretical measure of model performance in statistical machine learning – of multivariate linear regression and LAEs. We first propose a PAC-Bayes bound for multivariate linear regression, which is generalized from an earlier PAC-Bayes bound for single-output linear regression by Shalaeva et al., and outline sufficient conditions that ensure its theoretical convergence. We then apply this bound to EASE, a classic LAE model in recommender systems, and develop a practical method for minimizing the bound, addressing the calculation challenges posed by the zero-diagonal constraint. Experimental results show that our bound for EASE is non-vacuous on real-world datasets, demonstrating its practical utility.

## 1 Introduction

In recent years, simple (linear) recommendation models have consistently demonstrated impressive performance, often rivaling deep learning models (Dacrema et al., 2019; Jin et al., 2021; Mao et al., 2021), especially for the implicit setting, where interactions are inferred from user behavior (e.g., clicks or purchases). In particular, linear autoencoders (LAEs) such as EASE (Steck, 2019) and EDLAE (Steck, 2020) have shown a surprising edge over widely used matrix factorization (MF) methods such as ALS (Hu et al., 2008). The LAE architecture is remarkably simple: Let $R \in \mathbb{R}^{m \times n}$ be the data matrix and $W \in \mathbb{R}^{n \times n}$ be the parameter matrix, the LAE model is defined as $f_W(R) = RW$, where $W$ is trained to satisfy $f_W(R) \approx R$. $W$ is considered both an encoder and a decoder. Typically, we add constraints such as $\text{diag}(W) = 0$ to prevent $W$ from overfitting towards $I$ (Steck, 2019).

Despite their power and widespread use, linear autoencoders, particularly in the context of recommendation systems, remain theoretically underexplored. Recommendation research has understandably focused on performance evaluation to compare models, but issues such as weak baselines and unreliable sampled metrics often make these evaluations difficult to reproduce (Dacrema et al., 2019; Cremonesi & Jannach, 2021). A recent study attempted to provide a theoretical comparison between linear recommendation models, such as matrix factorization and LAE, using spectral analysis, showing that both approaches "reduce" the singular values of the original user-item data matrix $R$, albeit in different ways (Jin et al., 2021). Another related study investigates the loss landscape of low-rank LAEs, characterizing their critical points through the smooth submanifold theory (Kunin et al., 2019).

In this work, we aim to advance the theoretical understanding of linear autoencoder (LAE) models' generalizability using statistical learning theory. While generalization theory has been extensively studied for various machine learning and deep learning models (Vapnik, 1991; Dziugaite & Roy, 2017), its application to LAE recommendation models remains largely unexplored. To address this gap, we leverage PAC-Bayes theory (McAllester, 1998), which integrates the Probably Approximately Correct (PAC) framework with Bayesian inference. Our analysis produces a nonvacuous bound, offering practical insights into LAE performance on unseen data. It is worth noting that prior

work (Srebro et al., 2004) theoretically analyzed the generalizability of linear matrix factorization models, deriving a *vacuous* PAC bound based on covering numbers.

Our study to establish PAC-Bayes bounds for LAE models builds on the theoretical framework introduced by Shalaeva (Shalaeva et al., 2020), which provides a PAC-Bayes bound for multiple linear regression (a single dependent variable with multiple independent variables) under the assumption of Gaussian data. However, applying this framework to LAE models introduces several challenges:

1. **Multivariate Linear Regression**: The PAC-Bayes bound must be extended from the *multiple linear regression* setting to the *multivariate linear regression* scenario, which involves multiple dependent variables. Notably, PAC-Bayes bounds for multivariate linear regression – an important method and topic in statistical learning and inference – remain unexplored in the existing literature.

2. **Additional Convergence Requirements**: Our analysis reveals the need for additional convergence conditions beyond those presented in (Shalaeva et al., 2020). These conditions are essential for ensuring theoretical convergence in the more complex multivariate setting.

3. **Zero-Diagonal Constraint**: LAE models, such as EASE and EDLAE, enforce a structural zero-diagonal constraint on the weight matrix. This introduces unique theoretical challenges in adapting PAC-Bayes bounds from multivariate linear regression to LAE models.

This paper addresses the aforementioned challenges and makes the following key contributions:

• (Section 3) We develop a general theoretical PAC-Bayes bound for multivariate linear regression (Theorem 1), of which Shalaeva's bound (Shalaeva et al., 2020) for single-output multiple linear regression is a special case. Additionally, we propose sufficient conditions (Theorem 2) that guarantee convergence for both the new bound (Theorem 1) and Shalaeva's original bound (Shalaeva et al., 2020).

• (Section 4) We apply the bound of Theorem 1 to a LAE model for recommendation, EASE (Steck, 2019) and develop a practical method for calculating the optimal parameters that minimize the bound. Specifically, we incorporate the constraint $\text{diag}(W) = 0$ into the bound and resolve the calculation challenges that arise from it by presenting Theorem 3 and Theorem 4.

• (Section 5) We conduct experiments for the bound in Section 4 on real-world datasets, and the results show that the bound does not exceed $3\times$ of the test error on three out of four datasets we used.

• (Section 6) We conclude and discuss the empirical implication and potential application of PAC-Bayes bound for LAE models in recommendation setting.

All proofs of the theorems and lemmas presented in this paper are provided in Appendix A, while related works are discussed in Appendix D.

## 2 PRELIMINARIES

**Alquier's Bound (Alquier et al., 2016)**: Let $S = \{(x_i, y_i)\}_{i=1}^m$ be the dataset where $x_i \in \mathbb{R}^n$ is the feature vector and $y_i \in \mathbb{R}$ is the label. Suppose each $(x_i, y_i)$ is i.i.d. sampled from an unknown data distribution $\mathcal{D}$. Let $f_\theta : \mathbb{R}^n \to \mathbb{R}$ be the machine learning model where $\theta$ is the vector of parameters. Let $l$ be the loss function, $R^{\text{emp}}(\theta) = \frac{1}{m} \sum_{i=1}^m l(f_\theta(x_i), y_i)$ be the empirical risk and $R^{\text{true}}(\theta) = \mathbb{E}_{(x,y) \sim \mathcal{D}}[l(f_\theta(x), y)]$ be the true risk. Let $\pi$ be a prior distribution of $\theta$ and $\rho$ be the posterior distribution of $\theta$, then for any $\lambda > 0, \delta > 0$,

$$P\left(\mathbb{E}_{\theta \sim \rho}[R^{\text{true}}(\theta)] < \mathbb{E}_{\theta \sim \rho}[R^{\text{emp}}(\theta)] + \frac{1}{\lambda}\left[D(\rho \,||\, \pi) + \ln\frac{1}{\delta} + \Psi_{\pi,\mathcal{D},l}(\lambda, m)\right]\right) \geq 1 - \delta$$

where $\Psi_{\pi,\mathcal{D},l}(\lambda, m) = \ln \mathbb{E}_{\theta \sim \pi} \mathbb{E}_{S \sim \mathcal{D}^m}[e^{\lambda(R^{\text{true}}(\theta) - R^{\text{emp}}(\theta))}]$.

The PAC-Bayes bound has two types: empirical bound and oracle bound (Alquier, 2021). The oracle bound means the upper bound contains $R^{\text{true}}(W)$ and assumes $\mathcal{D}$ is given (only the oracle knows $\mathcal{D}$). Alquier's bound is an oracle bound. Shalaeva's bound is derived from Alquier's bound by assuming $f_\theta$ is a linear regression model and $\mathcal{D}$ is Gaussian distribution.

**Shalaeva's Bound (Shalaeva et al., 2020)**: In Alquier's bound, suppose $f_\theta(x) = \theta^T x$ where $\theta \in \mathbb{R}^n$. Assume $\mathcal{D}$ satisfies $x_i \sim \mathcal{N}(0, \sigma_x^2 I)$, and there exist $\theta^* \in \mathbb{R}^n$ such that $y_i = (\theta^*)^T x_i + e_i$ where

$e_i \sim \mathcal{N}(0, \sigma_e^2)$. Here $\sigma_x^2, \sigma_e^2$ are constants. Let the loss function be $l(f_\theta(x_i), y_i) = (\theta^T x_i - y_i)^2$, then

$$\Psi_{\pi, \mathcal{D}, l}(\lambda, m) = \ln \mathbb{E}_{\theta \sim \pi} \frac{\exp(\lambda v_\theta)}{(1 + \frac{\lambda v_\theta}{m/2})^{m/2}} \leq \ln \mathbb{E}_{\theta \sim \pi} \exp\left(\frac{2\lambda^2 v_\theta^2}{m}\right) \tag{1}$$

where $v_\theta = \sigma_x^2 \|\theta - \theta^*\|_2^2 + \sigma_e^2$.

**Convergence of Shalaeva's Bound**: The convergence analysis in (Shalaeva et al., 2020) is presented informally. Here we formally state their results as follows:

(1) Since $\lim_{m \to \infty}(1 + \frac{\lambda v_\theta}{m/2})^{m/2} = \exp(\lambda v_\theta)$, for any $\lambda > 0$, the term $\Psi_{\pi, \mathcal{D}, l}(\lambda, m)$ converges,

$$\lim_{m \to \infty} \Psi_{\pi, \mathcal{D}, l}(\lambda, m) = \lim_{m \to \infty} \ln \mathbb{E}_{\theta \sim \pi} \frac{\exp(\lambda v_\theta)}{(1 + \frac{\lambda v_\theta}{m/2})^{m/2}} = \ln \mathbb{E}_{\theta \sim \pi} \lim_{m \to \infty} \frac{\exp(\lambda v_\theta)}{(1 + \frac{\lambda v_\theta}{m/2})^{m/2}} = 0$$

(2) Let $d$ be a constant and $\lambda = m^{1/d}$, then $\ln \mathbb{E}_{\theta \sim \pi} \exp\left(\frac{2\lambda^2 v_\theta^2}{m}\right) = \ln \mathbb{E}_{\theta \sim \pi} \exp\left(2m^{2/d-1} v_\theta^2\right)$.

When $d > 2$, $\lim_{m \to \infty} m^{-1/d} \ln \mathbb{E}_{\theta \sim \pi} \exp\left(2m^{2/d-1} v_\theta^2\right) = 0$, thus the entire bound converges as $m \to \infty$.

$$\lim_{m \to \infty} \frac{1}{\lambda}\left[D(\rho \| \pi) + \ln \frac{1}{\delta} + \Psi_{\pi, \mathcal{D}, l}(\lambda, m)\right]$$

$$\leq \lim_{m \to \infty} m^{-1/d}\left[D(\rho \| \pi) + \ln \frac{1}{\delta}\right] + \lim_{m \to \infty} m^{-1/d} \ln \mathbb{E}_{\theta \sim \pi} \exp\left(2m^{2/d-1} v_\theta^2\right) = 0$$

Upon careful examination of their analysis, we found that additional conditions are needed to ensure the above convergency results, which were not discussed in their original paper. In (1), swapping $\lim$ and $\mathbb{E}$ is valid only under some specific conditions. For example, by *dominated convergence theorem* (Resnick, 1998), the condition can be $\mathbb{E}_{\theta \sim \pi}[\exp(\lambda v_\theta)] < \infty$. $\pi$ needs to be a distribution satisfying this condition. In (2), some choices of $\pi$ can cause divergence. For example, when $\pi$ is Gaussian distribution, we have $\ln \mathbb{E}_{\theta \sim \pi} \exp\left(2m^{2/d-1} v_\theta^2\right) = \infty$ for any $m > 0$, thus $\lim_{m \to \infty} m^{-1/d} \ln \mathbb{E}_{\theta \sim \pi} \exp\left(2m^{2/d-1} v_\theta^2\right) = \infty$ and the bound diverges. We will discuss these issues in Section 3.2.

**Multivariate Linear Regression (Johnson & Wichern, 2007)**: Let $S = \{(x_i, y_i)\}_{i=1}^m$ be the dataset where $x_i \in \mathbb{R}^n$ and $y_i \in \mathbb{R}^p$. Let $X = [x_1, x_2, ..., x_m] \in \mathbb{R}^{n \times m}$ be the input matrix, $Y = [y_1, y_2, ..., y_m] \in \mathbb{R}^{p \times m}$ be the target, $W \in \mathbb{R}^{p \times n}$ be the weight matrix of the linear model and $E = [e_1, e_2, ..., e_m] \in \mathbb{R}^{p \times m}$ be the error matrix. The linear regression is defined as

$$Y = WX + E$$

Usually we let the first dimension of every $x_i$ be 1, i.e., $X_{1*}$ is a vector of all 1s. We say the linear regression is *multivariate* if $p > 1$, and is *multiple* if $n > 2$.

We can apply a statistical assumption to the multivariate linear regression, where it is typically assumed that the errors $e_i$ and $e_j$ are independent for $i \neq j$, but the dimensions of each $e_i$ can be dependent. A common statistical assumption is shown in Assumption 1.

**EASE (LAE) Model (Steck, 2019)**: EASE is one of the most popular LAE models for recommendation (Jin et al., 2021). Let $R^{m \times n}$ be the data matrix and $W \in \mathbb{R}^{n \times n}$ be the weight matrix, then EASE obtains the model $W$ by solving the following problem

$$\min_W \|R - RW\|_F^2 + \gamma \|W\|_F^2 \quad \text{s.t. } \text{diag}(W) = 0 \tag{2}$$

where $\gamma$ is the regularization parameter. Let $W_0$ be the solution of Eq (2), then $W_0$ has closed from: Let $P = \left(R^T R + \gamma I\right)^{-1}$, then $(W_0)_{ij} = 0$ if $i = j$ and $(W_0)_{ij} = -P_{ij}/P_{jj}$ if $i \neq j$.

By structural risk minimization (Vapnik, 1991), the regularizer $\gamma \|W\|_F^2$ can be interpreted as a Lagrange multiplier term $\gamma(\|W\|_F^2 - c)$ for some constant $c$. Thus Eq (2) is equivalent to

$$\min_W \|R - RW\|_F^2 \quad \text{s.t. } \text{diag}(W) = 0, \|W\|_F^2 \leq c \tag{3}$$

Hence, tuning $\lambda$ in Eq (2) is equivalent to tuning $c$ in Eq (3), though the former form is more often used in practice. Note that by adding the constraint $\|W\|_F^2 \leq c$ we assume $\|W\|_F$ is bounded, which corresponds to case (1) and (3) in section 3.2.

# 3 PAC-Bayes Bound for Multivariate Linear Regression

## 3.1 The Statistical Assumption and the Bound

**Assumption 1** *Suppose each $(x_i, y_i)$ in $S$ is i.i.d. sampled from a distribution $\mathcal{D}$. $\mathcal{D}$ is defined as: (1) $x_i \sim \mathcal{N}(\mu_x, \Sigma_x)$; (2) there exist $W^* \in \mathbb{R}^{p \times n}$ and $e \sim \mathcal{N}(0, \Sigma_e)$ such that for any given $x_i$, $y_i = W^* x_i + e$, in other words, $y_i | x_i \sim \mathcal{N}(W^* x_i, \Sigma_e)$. Here $\mu_x \in \mathbb{R}^n$, $\Sigma_x \in \mathbb{R}^{n \times n}$ is positive semi-definite, and $\Sigma_e \in \mathbb{R}^{p \times p}$ is positive-definite.*

The positive semi-definite assumption of $\Sigma_x$ allows $\Sigma_x$ to be singular, implying that the Gaussian distribution is degenerate, i.e., its support is on a lower dimensional manifold embedded in $\mathbb{R}^n$. This includes the case that $x_i$ has its first dimension to be constant 1 and the other $n-1$ dimensions to be Gaussian random variables. In this case, the first row and first column of $\Sigma_x$ are 0.

Let $W \in \mathbb{R}^{p \times n}$ be the weight matrix of the linear model, then the prediction of the model on $x_i$ is given by $\hat{y}_i = W x_i$. The error is $y_i - \hat{y}_i = (W^* - W)x_i + e \sim \mathcal{N}(\mu_W, \Sigma_W)$, where

$$\mu_W = \mathbb{E}[(W^* - W)x_i + e] = (W^* - W)\mathbb{E}[x_i] + \mathbb{E}[e] = (W^* - W)\mu_x$$

$$\Sigma_W = \mathbb{E}[(W^* - W)(x_i - \mu_x) + e)][(W^* - W)(x_i - \mu_x) + e]^T$$

$$= (W^* - W)\Sigma_x(W^* - W)^T + \Sigma_e$$

It is easy to verify that $\Sigma_W$ is positive-definite. Thus, $\Sigma_W$ has an eigenvalue decomposition $\Sigma_W = S^T \Lambda S$ where $S$ is orthogonal, $\Lambda = \text{diag}(\eta_1, \eta_2, ..., \eta_p)$ and $\eta_i > 0$ for all $i$. Note that $S$ and $\Lambda$ depend on $W$.

Define the loss of the sample $(x_i, y_i)$ as $\|y_i - W x_i\|_F^2$, the empirical risk as $R^{\text{emp}}(W) = \frac{1}{m}\sum_{i=1}^m \|y_i - W x_i\|_F^2$ and the true risk as $R^{\text{true}}(W) = \mathbb{E}_{(x,y)\sim\mathcal{D}}[\|y - W x\|_F^2]$. Then we have the following bound:

**Theorem 1** *Let $\pi$ be the prior distribution of $W$, $\rho$ be the posterior distribution of $W$. Denote $b = S\Sigma_W^{-1/2}\mu_W$. Then for any $\lambda > 0$ and $\delta > 0$,*

$$P\left(\mathbb{E}_{W\sim\rho}[R^{\text{true}}(W)] < \mathbb{E}_{W\sim\rho}[R^{\text{emp}}(W)] + \frac{1}{\lambda}\left[D(\rho\,\|\,\pi) + \ln\frac{1}{\delta} + \Psi_{\pi,\mathcal{D}}(\lambda, m)\right]\right) \geq 1 - \delta \quad (4)$$

*where*

$$\Psi_{\pi,\mathcal{D}}(\lambda, m) = \ln\mathbb{E}_{W\sim\pi}\left[\exp\left(\lambda\left(\text{tr}(\Sigma_W) + \mu_W^T\mu_W\right)\right)\frac{\exp\left(\sum_{i=1}^p \frac{-\lambda m b_i^2 \eta_i}{m + 2\lambda\eta_i}\right)}{\prod_{i=1}^p (1 + 2\lambda\eta_i/m)^{m/2}}\right]$$

$$\leq \ln\mathbb{E}_{W\sim\pi}\exp\left(\frac{2\lambda^2\|\Sigma_W\|_F^2}{m}\right)$$

The bound of Theorem 1 is a general case of Shalaeva's bound. It can be reduced to Shalaeva's bound by taking $p = 1$, $\mu_x = 0$, $\Sigma_x = \sigma_x^2 I$ and $\Sigma_e = \sigma_e^2$ for some $\sigma_x, \sigma_e$.

## 3.2 Convergence Analysis

This section presents the convergence analysis of Theorem 1. We outline sufficient conditions that ensure convergence, thereby completing and rigorously formalizing the convergence analysis of Shalaeva's bound (Shalaeva et al., 2020)

We first discuss the convergence of $\Psi_{\pi,\mathcal{D}}(\lambda, m)$ term, then the entire bound. Theorem 2 gives a sufficient condition for the convergence of $\Psi_{\pi,\mathcal{D}}(\lambda, m)$ based on the dominated convergence theorem.

**Theorem 2** *If $\lambda$ and $\pi$ satisfies $\mathbb{E}_{W\sim\pi}\left[\exp\left(\lambda\|(\Sigma_x + \mu_x\mu_x^T)^{1/2}(W^* - W)\|_F^2\right)\right] < \infty$, then $\lim_{m\to\infty}\Psi_{\pi,\mathcal{D}}(\lambda, m) = 0$.*

By Theorem 2, we can derive some special cases that make $\Psi_{\pi,\mathcal{D}}(\lambda, m)$ converge:

(1) If $\pi$ is a bounded distribution such that $\|W\|_F < G$ where $G$ is a constant, then for any $\lambda > 0$,

$$\mathbb{E}_{W\sim\pi}\left[\exp\left(\lambda\|(\Sigma_x + \mu_x\mu_x^T)^{1/2}(W^* - W)\|_F^2\right)\right] \leq \mathbb{E}_{W\sim\pi}\left[\exp\left(\lambda\|(\Sigma_x + \mu_x\mu_x^T)^{1/2}\|_F^2\|W^* - W\|_F^2\right)\right]$$

$$\leq \exp\left(\lambda\|(\Sigma_x + \mu_x\mu_x^T)^{1/2}\|_F^2\left(\|W^*\|_F + \|W\|_F\right)^2\right) < \exp\left(\lambda\|(\Sigma_x + \mu_x\mu_x^T)^{1/2}\|_F^2\left(\|W^*\|_F + G\right)^2\right) < \infty$$

(2) If $\pi$ is a distribution that for $W \sim \pi$, each $W_{ij}$ is independently sampled from $\mathcal{N}((\mathcal{U}_0)_{ij}, \sigma^2)$ where $\sigma > 0$ is a constant and $\mathcal{U}_0 \in \mathbb{R}^{n \times n}$. Then for any $\lambda \in (0, \frac{1}{2\eta_1\sigma^2})$, $\mathbb{E}_{W \sim \pi}\left[\exp\left(\lambda\|(\Sigma_x + \mu_x\mu_x^T)^{1/2}(W^* - W)\|_F^2\right)\right] < \infty$ holds. This is because, let $\Sigma_x + \mu_x\mu_x^T = S^T\Lambda S$ be the eigenvalue decomposition and suppose $\Lambda = \text{diag}(\eta_1, \eta_2, ..., \eta_n)$ where $\eta_1$ is the largest eigenvalue, then

$$\mathbb{E}_{W \sim \pi}\left[\exp\left(\lambda\|(\Sigma_x + \mu_x\mu_x^T)^{1/2}(W^* - W)\|_F^2\right)\right] = \prod_{i=1}^{p}\prod_{j=1}^{p}\frac{\exp\left(\frac{\lambda\eta_j\left(S_{j*}(W^*-\mathcal{U}_0)_{*i}\right)^2}{1-2\lambda\sigma^2\eta_j}\right)}{(1-2\lambda\sigma^2\eta_j)^{1/2}}$$

And $\lambda \in (0, \frac{1}{2\eta_1\sigma^2})$ ensures denominator $\left(1 - 2\lambda\sigma^2\eta_j\right)^{1/2}$ is not zero or undefined for any $j$.

Now we discuss the convergence of the entire bound when $\lambda = m^{1/d}$. Since $\frac{1}{\lambda}\left[D(\rho\,\|\,\pi) + \ln\frac{1}{\delta}\right]$ surely converges as $m \to \infty$, we only discuss the convergence of $\frac{1}{\lambda}\Psi_{\pi,\mathcal{D}}(\lambda, m)$. By Theorem 1, $\frac{1}{\lambda}\Psi_{\pi,\mathcal{D}}(\lambda, m)$ converges if the upper bound $\frac{1}{\lambda}\ln\mathbb{E}_{W \sim \pi}\exp\left(\frac{2\lambda^2\|\Sigma_W\|_F^2}{m}\right)$ converges.

(3) If $\pi$ is a bounded distribution satisfying $\|W\|_F < G$, then

$$\|\Sigma_W\|_F^2 = \|(W^* - W)\Sigma_x(W^* - W)^T + \Sigma_e\|_F^2 \leq \left(\|(W^* - W)\Sigma_x(W^* - W)^T\|_F + \|\Sigma_e\|_F\right)^2$$

$$\leq \left(\|\Sigma_x\|_F\|W^* - W\|_F^2 + \|\Sigma_e\|_F\right)^2 \leq \left(\|\Sigma_x\|_F\left(\|W^*\|_F + \|W\|_F\right)^2 + \|\Sigma_e\|_F\right)^2$$

$$< \left(\|\Sigma_x\|_F\left(\|W^*\|_F + G\right)^2 + \|\Sigma_e\|_F\right)^2 < \infty$$

Denote $G' = \left(\|\Sigma_x\|_F\left(\|W^*\|_F + G\right)^2 + \|\Sigma_e\|_F\right)^2$. The upper bound converges when $d > 2$:

$$\lim_{m \to \infty} m^{-1/d}\ln\mathbb{E}_{W \sim \pi}\exp\left(2m^{2/d-1}\|\Sigma_W\|_F^2\right) < \lim_{m \to \infty} m^{-1/d}\ln\mathbb{E}_{W \sim \pi}\exp\left(2m^{2/d-1}G'\right) = 0$$

(4) If $\pi$ is a distribution that for $W \sim \pi$, each $W_{ij}$ is a Gaussian random variable, then the upper bound diverges when $d > 2$, thus we cannot show the convergence of $\frac{1}{\lambda}\Psi_{\pi,\mathcal{D}}(\lambda, m)$. We prove the divergence of the upper bound as follows. First, for any $r, q \in \{1, 2, ..., p\}$,

$$\|\Sigma_W\|_F^2 = \sum_{i=1}^{p}\sum_{j=1}^{p}\left((W^* - W)_{*i}^T\Sigma(W^* - W)_{*j} + (\Sigma_e)_{ij}\right)^2$$

$$\geq \left((W^* - W)_{*q}^T\Sigma_x(W^* - W)_{*q} + (\Sigma_e)_{qq}\right)^2 = \left(\|(\Sigma_x)^{1/2}(W^* - W)_{*q}\|_2^2 + (\Sigma_e)_{qq}\right)^2$$

$$\geq \left(\|(\Sigma_x)^{1/2}(W^* - W)_{*q}\|_2^2\right)^2 \geq \left((\Sigma_x)_{r*}^{1/2}(W^* - W)_{*q}\right)^4$$

In the above inequality we use the fact that $(\Sigma_x)_{qq} \geq 0$ since it is a diagonal element of $\Sigma_x$. Since $(W^* - W)_{*q}$ is a random Gaussian vector, $(\Sigma)_{r*}^{1/2}(W^* - W)_{*q}$ is a Gaussian random variable. Denote $w = (\Sigma)_{r*}^{1/2}(W^* - W)_{*q}$, then

$$m^{-1/d}\ln\mathbb{E}_{W \sim \pi}\exp\left(2m^{2/d-1}\|\Sigma_W\|_F^2\right) \geq m^{-1/d}\ln\mathbb{E}_w\exp\left(2m^{2/d-1}w^4\right)$$

**Lemma 1** *Let $\{a_k\}_{i=0}^{k}$ be a sequence of real numbers. Let $X$ be a Gaussian random variable and $Y_k = \sum_{i=0}^{k} a_i X^i$ where $a_k > 0$. If $k \geq 3$, then $Y_k$ has no MGF, i.e., $M_{Y_k}(t) = \mathbb{E}_{Y_k}[\exp(tY_k)] = \mathbb{E}_X[\exp(tY_k)] = \infty$ for any $t > 0$.*

Lemma 1 states that any polynomial of Gaussian random variables of degree $\geq 3$ has no MGF. The term $w^4$ satisfies the conditions of Lemma 1 as a polynomial of degree 4. Thus we have $\mathbb{E}_w\exp\left(2m^{2/d-1}w^4\right) = \infty$ for any $m > 0$, and $\ln\mathbb{E}_w\exp\left(2m^{2/d-1}w^4\right) = \infty$. Note that when $m \to \infty$, $m^{-1/d}$ and $m^{2/d-1}$ are positive numbers being arbitrary close to 0 but never equivalent to 0. Thus $\lim_{m \to \infty} m^{-1/d}\ln\mathbb{E}_w\exp\left(2m^{2/d-1}w^4\right) = \infty$. This shows the upper bound diverges.

Recall that Shalaeva's bound in Section 2 has $v_\theta = \sigma_x^2\|\theta - \theta^*\|_2^2 + \sigma_e^2$. When $\theta$ is a Gaussian vector, $v_\theta^2$ becomes a polynomial of Gaussian random variables of degree 4, which satisfies the condition of Lemma 1. Thus the divergence $\lim_{m \to \infty}\ln\mathbb{E}_{\theta \sim \pi}\exp\left(2m^{2/d-1}v_\theta^2\right) = \infty$ cannot be resolved by taking any $d > 2$.

# 4 A PRACTICAL PAC-BAYES BOUND FOR LAE

This section introduces how to apply the bound of Theorem 1 to EASE, a simple yet very effective LAE recommendation model, and provides a practical way to calculate the bound.

## 4.1 THE SETTINGS AND THE BOUND

The EASE model can be considered as a special case of multivariate linear regression, where $Y$ is equivalent to $X$ and $W$ is constrained by $\text{diag}(W) = 0$. Also in recommender system, the dataset $R$ is usually not Gaussian but bounded. To apply the bound of Theorem 1 to EASE, we redefine our settings as follows:

Suppose each $R_i^T$ in $R$ is i.i.d. sampled from an unknown $n$ dimensional bounded distribution $\mathcal{D}$. Also, assume $\mathcal{D}$ is a bounded distribution satisfying the condition that there exists $a, b$ such that $R_{ij} \in [a, b]$ for any $i, j$. Define the loss function on $R_i$ as $\|R_i - R_i W\|_F^2$, the empirical risk as $R^{\text{emp}}(W) = \frac{1}{m} \|R - RW\|_F^2 = \frac{1}{m} \sum_{i=1}^{m} \|R_i - R_i W\|_F^2$, and the true risk as $R^{\text{true}}(W) = \mathbb{E}_{r \sim \mathcal{D}}[\|r^T - r^T W\|_F^2]$.

Then the PAC-Bayes bound for EASE is as follows (the same form as Eq (2) but with different settings):

$$P\left( \mathbb{E}_{W \sim \rho}[R^{\text{true}}(W)] < \mathbb{E}_{W \sim \rho}[R^{\text{emp}}(W)] + \underbrace{\frac{1}{\lambda} D(\rho \,\|\, \pi) + \frac{1}{\lambda} \ln \frac{1}{\delta}}_{\text{part 1}} + \underbrace{\frac{1}{\lambda} \Psi_{\pi, \mathcal{D}}(\lambda, m)}_{\text{part 2}} \right) \geq 1 - \delta \quad (5)$$

with $R^{\text{emp}}(W) = \frac{1}{m} \|R - RW\|_F^2$, $R^{\text{true}}(W) = \mathbb{E}_{r \sim \mathcal{D}}[\|r^T - r^T W\|_F^2]$, $\text{diag}(W) = 0$, and $\Psi_{\pi, \mathcal{D}}(\lambda, m) = \ln \mathbb{E}_{W \sim \pi} \mathbb{E}_{R \sim \mathcal{D}^m} \left[ e^{\lambda(R^{\text{true}}(W) - R^{\text{emp}}(W))} \right]$.

We aim to find a practical method for calculating the tightest bound, so that it can provide theoretical support for practical applications. For any given $\delta$, our goal is to find $\lambda, \pi, \rho$ that minimizes the right hand side of Eq (5). It is generally considered difficult to solve for $\lambda, \pi, \rho$ simultaneously (Alquier, 2021), so we typically fix $\lambda, \pi$ and solve for $\rho$. We show how to minimize part 1 of Eq (5) in Section 4.2 and how to find a practical upper bound for part 2 of Eq (5) in Section 4.3.

## 4.2 CLOSED-FORM SOLUTION FOR THE OPTIMAL $\rho$

Since the PAC-Bayes bound holds for any $\pi, \rho$ and $\lambda$, given $\pi$ and $\lambda$, we search for the optimal $\rho$ by

$$\min_{\rho} \mathbb{E}_{W \sim \rho}[R^{\text{emp}}(W)] + \frac{1}{\lambda} D(\rho \,\|\, \pi) \quad (6)$$

Usually we restrict $\pi$ and $\rho$ to be specific distributions that make Eq (6) easy to calculate. (Dziugaite & Roy, 2017) proposed a practical way to calculate the PAC-Bayes bound for deep neural networks, where they assumes $\pi$ and $\rho$ to be independent multivariate Gaussian. This enables the $D(\rho \,\|\, \pi)$ term to be easily calculated. We mainly follow the assumptions in (Dziugaite & Roy, 2017):

**Assumption 2** *Denote $\bar{\mathcal{N}}(\mathcal{A}, \mathcal{B})$ for some $\mathcal{A} \in \mathbb{R}^{n \times n}$ and non-negative $\mathcal{B} \in \mathbb{R}^{n \times n}$ as the multivariate Gaussian distribution that $W \sim \bar{\mathcal{N}}(\mathcal{A}, \mathcal{B})$ means $W \in \mathbb{R}^{n \times n}$ and each $W_{ij}$ is independently from $\mathcal{N}(\mathcal{A}_{ij}, \mathcal{B}_{ij})$. Assume $\rho$ is the distribution $\bar{\mathcal{N}}(\mathcal{U}, \mathcal{S})$ and $\pi$ is the distribution $\bar{\mathcal{N}}(\mathcal{U}_0, \sigma^2 J)$, where $\mathcal{U} \in \mathbb{R}^{n \times n}$, $\mathcal{U}_0 \in \mathbb{R}^{n \times n}$, $\mathcal{S} \in \mathbb{R}^{n \times n}$, $J = \{1\}^{n \times n}$ and $\sigma > 0$. $\mathcal{S}$ is a positive matrix if no constraint is applied.*

Applying the constraint $\text{diag}(W) = 0$ to $\rho$ and $\pi$ is equivalent to set $\text{diag}(\mathcal{U}) = 0, \text{diag}(\mathcal{S}) = 0$, $\text{diag}(\mathcal{U}_0) = 0$ and $\text{diag}(\sigma^2 J) = 0$.

(Dziugaite & Roy, 2017) solved the optimal $\rho$ using stochastic gradient descent, where in each iteration the gradient is calculated by Monte Carlo method. It should be noticed that Dziugaite and Roy used the iterative method because they worked on the neural network model, for which the optimal $\rho$ may not have a closed-form solution. Due to the simplicity of LAE, we find that the optimal $\rho$ for Eq (6) has closed-form solution, as shown in Theorem 3 (1). This allows us to solve $\rho$ directly and avoid time-consuming iterative methods.

**Theorem 3** *(1) The closed-form solution of the optimal $\rho$ of Eq (6) is given by*

$$\mathcal{U} = \left(\frac{1}{m}R^T R + \frac{1}{2\lambda\sigma^2}I\right)^{-1}\left(\frac{1}{m}R^T R + \frac{1}{2\lambda\sigma^2}\mathcal{U}_0\right), \quad \mathcal{S}_{ij} = \frac{1}{\frac{2\lambda}{m}R_{*i}^T R_{*i} + \frac{1}{\sigma^2}} \quad for \ i,j \in \{1,2,...,n\}$$

*(2) If we add the constraint $\mathrm{diag}(W) = 0$ to $\rho$ and $\pi$, then the optimal $\rho$ becomes*

$$\mathcal{S}_{ij} = \frac{1}{\frac{2\lambda}{m}R_{*i}^T R_{*i} + \frac{1}{\sigma^2}}, \ \mathcal{S}_{ii} = 0 \ \ for \ i,j \in \{1,2,...,n\} \ and \ i \neq j$$

$$\mathcal{U} = \left(\frac{1}{m}R^T R + \frac{1}{2\lambda\sigma^2}I\right)^{-1}\left(\frac{1}{m}R^T R + \frac{1}{2\lambda\sigma^2}\mathcal{U}_0 - \frac{1}{2}\mathrm{Diag}(x)\right)$$

*where*

$$x = 2 \cdot \mathrm{diag}\left[\left(\frac{1}{m}R^T R + \frac{1}{2\lambda\sigma^2}I\right)^{-1}\left(\frac{1}{m}R^T R + \frac{1}{2\lambda\sigma^2}\mathcal{U}_0\right)\right] \oslash \mathrm{diag}\left[\left(\frac{1}{m}R^T R + \frac{1}{2\lambda\sigma^2}I\right)^{-1}\right]$$

*Here $\oslash$ means element-wise division and $\mathrm{Diag}(x)$ means expanding $x \in \mathbb{R}^n$ to an $n \times n$ diagonal matrix.*

Once the $\mathcal{U}$ and $\mathcal{S}$ for the optimal $\rho$ are obtained, we can calculate the closed-form solutions of $\mathbb{E}_{W\sim\rho}[R^{\mathrm{emp}}(W)]$ and $D(\rho\,||\,\pi)$, as shown in the proof of Theorem 3.

### 4.3 Easy-to-calculate Upper Bound for $\Psi_{\pi,\mathcal{D}}(\lambda, m)$

Since $\Psi_{\pi,\mathcal{D}}(\lambda, m) = \ln\mathbb{E}_\pi\mathbb{E}_\mathcal{D}[e^{\lambda\left(R^{\mathrm{true}}(W)-R^{\mathrm{emp}}(W)\right)}]$ and $R^{\mathrm{emp}}(W) \geq 0$, based on the idea of (Germain et al., 2016), we can get an upper bound of $\Psi$ by removing $-R^{\mathrm{emp}}(W)$: Let $\Psi'_{\pi,\mathcal{D}}(\lambda) = \ln\mathbb{E}_\pi[e^{\lambda R^{\mathrm{true}}(W)}]$, then $\Psi_{\pi,\mathcal{D}}(\lambda, m) \leq \Psi'_{\pi,\mathcal{D}}(\lambda)$. $\Psi'$ does not converge as $m \to \infty$ since it is independent of $m$, but it is easier to calculate than $\Psi$.

Denote $\Sigma_r = \mathbb{E}_{r\sim\mathcal{D}}[rr^T]$, then

$$\mathbb{E}_\pi\left[e^{\lambda R^{\mathrm{true}}(W)}\right] = \mathbb{E}_\pi[\exp\lambda\mathbb{E}_{r\sim\mathcal{D}}[r^T(I-W)]] = \mathbb{E}_\pi\left[\exp\lambda\left(\sum_{i=1}^n (I-W)_{*i}^T\Sigma_r(I-W)_{*i}\right)\right]$$

$$= \mathbb{E}_\pi\left[\exp\lambda\left(\sum_{i=1}^n\left\|\Sigma_r^{1/2}(I-W)_{*i}\right\|_F^2\right)\right] = \prod_{i=1}^n\mathbb{E}_\pi\left[\exp\lambda\left(\left\|\Sigma_r^{1/2}(I-W)_{*i}\right\|_F^2\right)\right]$$

Since $(I-W)_{ii} = 0$, $(I-W)_{*i} \sim \mathcal{N}\left((I-\mathcal{U}_0)_{*i}, \sigma^2(I-I^i)\right)$, where $I^i$ is a matrix with $I_{ii}^i = 1$ and other entries being 0. So $\Sigma_r^{1/2}(I-W)_{*i} \sim \mathcal{N}\left(\Sigma_r^{1/2}(I-\mathcal{U}_0)_{*i}, \sigma^2(\Sigma_r - (\Sigma_r^{1/2})_{*i}(\Sigma_r^{1/2})_{*i}^T)\right)$.

Denote $A^{(i)} = \sigma^2(\Sigma_r - (\Sigma_r^{1/2})_{*i}(\Sigma_r^{1/2})_{*i}^T)$, then $A^{(i)}$ is singular and positive semi-definite. Let $A^{(i)} = S^{(i)T}\Lambda^{(i)}S^{(i)}$ be the eigenvalue decomposition where $S^{(i)}$ is orthogonal and $\Lambda^{(i)} = \mathrm{diag}(\eta_1^{(i)}, \eta_2^{(i)}, ..., \eta_n^{(i)})$. Also denote $\mu^i = \Sigma_r^{1/2}(I-\mathcal{U}_0)_{*i}$. Then

$$\mathbb{E}_\pi\left[e^{\lambda R^{\mathrm{true}}(W)}\right] = \prod_{i=1}^n\prod_{j=1}^n \frac{\exp\left(\frac{\lambda(b_j^{(i)})^2\eta_j^{(i)}}{1-2\lambda\eta_j}\right)}{\left(1-2\lambda\eta_j^{(i)}\right)^{1/2}}, \quad \text{where } b^{(i)} = S^{(i)}(A^{(i)})^{-1/2}\mu^i \qquad (7)$$

Eq (7) is obtained by applying Eq (11) where we take $m = 1$. The problem with Eq (7) is the computational complexity: We need to calculate the eigenvalue decomposition for each $A^{(i)}$ in order to obtain $S^{(i)}$ and $\Lambda^{(i)}$. Since each eigenvalue decomposition costs $O(n^3)$, the computation of Eq (7) costs $O(n^4)$, which is impractical.

Here we show how to find a practical upper bound for Eq (7). Let $\pi'$ be the distribution that for any $W \sim \pi'$, $\mathrm{diag}(W) \sim \mathcal{N}(0, \sigma^2 I)$. The only difference between $\pi$ and $\pi'$ is that, $\pi$ constrains

diag$(W)$ to be constant zeros, while $\pi'$ constrains the diag$(W)$ to be i.i.d. Gaussian random variables with zero mean.

$W \sim \pi'$ gives $(I - W)_{*i} \sim \mathcal{N}\left((I - \mathcal{U}_0)_{*i}, \sigma^2 I\right)$, $\Sigma_r^{1/2}(I - W)_{*i} \sim \mathcal{N}\left(\Sigma_r^{1/2}(I - \mathcal{U}_0)_{*i}, \sigma^2 \Sigma_r\right)$.
Let $A = \sigma^2 \Sigma_r$, and $A = S^T \Lambda S$ be the eigenvalue decomposition where $S$ is orthogonal and $\Lambda = \mathrm{diag}(\eta_1, \eta_2, ..., \eta_n)$, then we have

$$\mathbb{E}_{\pi'}\left[e^{\lambda R^{\mathrm{true}}(W)}\right] = \prod_{i=1}^{n}\prod_{j=1}^{n} \frac{\exp\left(\frac{\lambda(\bar{b}_j^i)^2 \eta_j}{1 - 2\lambda\eta_j}\right)}{(1 - 2\lambda\eta_j)^{1/2}}, \quad \text{where } \bar{b}^i = SA^{-1/2}\mu^i \tag{8}$$

and the following theorem:

**Theorem 4** $\mathbb{E}_{\pi}\left[e^{\lambda R^{\mathrm{true}}(W)}\right] \leq \mathbb{E}_{\pi'}\left[e^{\lambda R^{\mathrm{true}}(W)}\right]$ *for any* $\lambda \in \left(0, \frac{1}{2\eta_1}\right)$.

Note that $\mathbb{E}_{\pi'}\left[e^{\lambda R^{\mathrm{true}}(W)}\right]$ is much easier to calculate: We only need to calculate the eigenvalue decomposition of $A$, so Eq (8) costs $O(n^3)$. Let $\Psi'_{\pi',\mathcal{D}}(\lambda) = \ln \mathbb{E}_{\pi'}\left[e^{\lambda R^{\mathrm{true}}(W)}\right]$, then $\Psi'_{\pi',\mathcal{D}}(\lambda) \geq \Psi'_{\pi,\mathcal{D}}(\lambda)$. Hence $\Psi'_{\pi',\mathcal{D}}(\lambda)$ is a practical upper bound for $\Psi'_{\pi,\mathcal{D}}(\lambda)$.

The last thing we need to do is to obtain $\Sigma_r$. Since $\mathcal{D}$ is unknown, we can not calculate $\Sigma_r$ directly, thus we need an approximation.

Let $R' \in \mathbb{R}^{m' \times n}$ be the entire dataset where we take the first $m$ rows to be the training set and the rest $m' - m$ rows to be the test set, and let $\hat{\Sigma}_r = \frac{1}{m'} R'^T R'$, then $\hat{\Sigma}_r$ is an unbiased estimator of $\Sigma_r$. This is because, let $\mathcal{M}$ be a distribution such that $r \sim \mathcal{D}$ is equivalent to $rr^T \sim \mathcal{M}$, then each $R'^T_{i*} R'_{i*}$ is i.i.d. sampled from $\mathcal{M}$, $\hat{\Sigma}_r$ is the sample mean, and $\Sigma_r$ is the expectation. By law of large numbers, we have $\hat{\Sigma}_r \xrightarrow{p} \Sigma_r$ as $m' \to \infty$. Therefore, we use $\hat{\Sigma}_r$ to approximate $\Sigma_r$. The error between $\hat{\Sigma}_r$ to approximate $\Sigma_r$ is discussed in Appendix C.

Let $\hat{\Sigma}_r = S'^T \Lambda' S'$ be the eigenvalue decomposition where $\Lambda' = \mathrm{diag}(\eta'_1, \eta'_2, ..., \eta'_n)$. The approximation of Eq (8) can be made by replacing $\Sigma_r$ with $\hat{\Sigma}_r$, specifically, replacing $\bar{b}^i$ with $\frac{1}{\sigma}S'(I - \mathcal{U}_0)_{*i}$ and $\eta_j$ with $\sigma^2 \eta'_j$ for all $j$ in Eq (8). Then

$$\hat{\mathbb{E}}_{\pi'}\left[e^{\lambda R^{\mathrm{true}}(W)}\right] = \prod_{i=1}^{n}\prod_{j=1}^{n} \frac{\exp\left(\frac{\lambda\eta'_j\left(S'_{j*}(I - \mathcal{U}_0)_{*i}\right)^2}{1 - 2\lambda\sigma^2\eta'_j}\right)}{\left(1 - 2\lambda\sigma^2\eta'_j\right)^{1/2}} = \prod_{j=1}^{n} \frac{\exp\left(\frac{\lambda\eta'_j\left\|S'_{j*}(I - \mathcal{U}_0)\right\|_F^2}{1 - 2\lambda\sigma^2\eta'_j}\right)}{\left(1 - 2\lambda\sigma^2\eta'_j\right)^{n/2}} \approx \mathbb{E}_{\pi'}\left[e^{\lambda R^{\mathrm{true}}(W)}\right]$$

Denote $\mathrm{approx}(\Psi'_{\pi',\mathcal{D}}(\lambda)) = \ln \hat{\mathbb{E}}_{\pi'}\left[e^{\lambda R^{\mathrm{true}}(W)}\right]$, then $\mathrm{approx}(\Psi'_{\pi',\mathcal{D}}(\lambda)) \approx \Psi'_{\pi',\mathcal{D}}(\lambda)$.

### 4.4 THE FINAL BOUND

Consider we fix $\pi$ and search $\lambda$ in a set $L = \{\lambda_1, \lambda_2, ...\lambda_l\}$. If the set contains $|L|$ candidate values for $\lambda$, then the term $|L|$ should be included to the bound. See Appendix B.

The final bound is shown as follows: Let $\mathcal{U}_0, \sigma$ be the parameters of $\pi$ and $\mathcal{U}, \mathcal{S}$ be the parameters of $\rho$. Suppose $\mathcal{U}_0, \sigma$ are given. For any $\lambda \in L$, with probability at least $1 - \delta$,

$$\mathbb{E}_{W \sim \rho}[R^{\mathrm{true}}(W)] \lessapprox \mathbb{E}_{W \sim \rho}[R^{\mathrm{emp}}(W)] + \frac{1}{\lambda}\left[D(\rho \,||\, \pi) + \ln\frac{|L|}{\delta} + \mathrm{approx}\left(\Psi'_{\pi',\mathcal{D}}(\lambda)\right)\right] \tag{9}$$

where

$$\mathbb{E}_{W \sim \rho}[R^{\mathrm{emp}}(W)] = \frac{1}{m}\sum_{i=1}^{m} R_i V R_i^T = \|V^{1/2}R\|_F^2$$

$$V = I - \mathcal{U} - \mathcal{U}^T + \mathcal{U}\mathcal{U}^T + \mathrm{diag}\left(\sum_{j=1}^{n}\mathcal{S}_{1j}, \sum_{j=1}^{n}\mathcal{S}_{2j}, ..., \sum_{j=1}^{n}\mathcal{S}_{nj}\right)$$

$$D(\rho \,||\, \pi) = \frac{1}{2}\left[(n^2 - n)(2\ln\sigma - 1) - \sum_{i=1}^{n}\sum_{j=1, j\neq i}^{n}\left(\ln\mathcal{S}_{ij} - \frac{\mathcal{S}_{ij}}{\sigma^2}\right) + \frac{\|\mathcal{U} - \mathcal{U}_0\|_F^2}{\sigma^2}\right]$$

$$\text{approx}\left(\Psi'_{\pi',\mathcal{D}}(\lambda)\right) = \sum_{i=1}^{n}\left(\frac{\lambda\eta'_i}{1-2\lambda\eta'_i\sigma^2}\left\|S'_{i*}(I-\mathcal{U}_0)\right\|_F^2 - \frac{n}{2}\ln\left(1-2\lambda\eta'_i\sigma^2\right)\right)$$

The whole process to calculate the PAC-Bayes bound for EASE is summarized as Algorithm 1.

---

**Algorithm 1** Calculate the PAC-Bayes bound for EASE

---

Initialize $L = \{\lambda_1, \lambda_2, ..., \lambda_l\}$, $\delta$, $\mathcal{U}_0$, $\sigma$, and an empty array $A = \{\}$.
**for** each $\lambda_i$ **in** $L$:
    Calculate $\mathcal{U}, \mathcal{S}$ by Theorem 3 (2).
    Calculate the right hand side of Eq (9), store the result as $A_i$, and append $A_i$ to $A$.
**return** the minimum element in $A$.

---

## 5 EXPERIMENTS

Our experiments run on a machine with 500 GB RAM and a Nvidia A100 GPU. The GPU has 80 GB RAM. We use 4 datasets: MovieLens 20M, Netflix, Yelp2018 and MSD. The details of the datasets are shown in Table 1.

Table 1: Dataset information

| Dataset | MovieLens 20M | Netflix | Yelp2018 | MSD |
|---|---|---|---|---|
| #rows | 138493 | 480189 | 905136 | 1017982 |
| #rows (training set) | 117718 | 408160 | 769365 | 865284 |
| #columns | 26744 | 17770 | 40000 | 40000 |
| #ratings | 2000263 | 100480507 | 1969320 | 33687193 |
| rating range | $[0, 5]$ | $[0, 5]$ | $[0, 5]$ | $[0, 9667]$ |

For Yelp2018 and MSD datasets, we truncated the rating matrices by keeping the first 40000 columns and all the rows containing non-zero elements in the first 40000 columns. For each dataset, we take the first $85\%$ of rows of the rating matrix as the training set and the rest $15\%$ rows as the test set.

The computation of PAC-Bayes bound for EASE mainly follows Algorithm 1. We set $L = \{1, 2, 4, 8, 16, 32, 64, 128, 256, 512\}$, $\delta = 0.01$, $\sigma = 0.001$. For each dataset and each choice of $\gamma$ in the set $\{50, 100, 200, 400\}$, we solve the $W_0$ of Eq (2), set $\mathcal{U}_0 = W_0$, and run Algorithm 1 to calculate the PAC-Bayes bound.

We evaluate the non-vacuousness by comparing the gap between the PAC-Bayes bound and the test error. To the best of our knowledge, there is no universally accepted definition for how small the gap must be to consider a theoretical bound non-vacuous. (Dziugaite & Roy, 2017) showed in their experiments that PAC-Bayes bounds within $10\times$ the test error can be considered non-vacuous. We adopt this criterion in our work.

The results are shown in Table 2. Our PAC-Bayes bound is within $3\times$ the test error on MovieLens 20M, Netflix and MSD, and is within $4\times$ the test error on Yelp2018, for all choices of $\gamma$. Thus we consider the bound non-vacuous.

Since the bound is composed of the terms $\lambda$, $D(\rho\,\|\,\pi)$, $R^{\text{emp}}(W)$ and approx $\left(\Psi'_{\pi',\mathcal{D}}(\lambda)\right)$, for each PAC-Bayes bound result in Table 2, we we present the corresponding values of these terms in Table 3 of Appendix E.

## 6 CONCLUSIONS AND DISCUSSIONS

This paper studies the generalizability of multivariate linear regression and LAE. We propose a new PAC-Bayes bound for multivariate linear regression, which generalizes Shalaeva's bound for multiple linear regression (Shalaeva et al., 2020). We also present a convergence analysis and demonstrate the sufficient conditions that ensure the bound's convergence. To illustrate how the bound applies to LAE, we use it with EASE, a simple yet very effective LAE recommendation model, and develop a practical method to calculate the optimal parameters that minimize the bound. This method primarily addresses the calculation challenges introduced by the zero diagonal constraint of EASE.

Table 2: Experiment results of the PAC-Bayes bound for EASE

| Dataset | | MovieLens 20M | Netflix | Yelp2018 | MSD |
|---|---|---|---|---|---|
| $\gamma = 50$ | training error | 737.54 | 1359.39 | 33.18 | 1172.46 |
| | test error | 1368.78 | 1661.67 | 18.29 | 1965.40 |
| | **PAC-Bayes bound** | **1674.76** | **2870.11** | **61.50** | **2436.74** |
| $\gamma = 100$ | training error | 728.59 | 1277.50 | 33.52 | 1174.46 |
| | test error | 1290.19 | 1627.43 | 17.93 | 1946.83 |
| | **PAC-Bayes bound** | **1696.18** | **2870.23** | **61.16** | **2433.95** |
| $\gamma = 200$ | training error | 774.94 | 1362.24 | 34.02 | 1177.73 |
| | test error | 1240.14 | 1638.99 | 17.60 | 1923.70 |
| | **PAC-Bayes bound** | **1724.65** | **2871.98** | **60.74** | **2435.44** |
| $\gamma = 400$ | training error | 797.71 | 1366.47 | 34.66 | 1182.98 |
| | test error | 1193.81 | 1622.19 | 17.32 | 1895.80 |
| | **PAC-Bayes bound** | **1759.83** | **2877.29** | **60.25** | **2438.74** |

**Extending to other LAE models:** Another class of Linear Autoencoder (LAE) models employs low-rank approximations to represent and constrain $W$. While our multivariate linear regression approach can potentially be applied and generalized to these models (though special handling is needed to model a low-rank $W$ from a certain distribution, which is non-trivial), they are generally less effective than the zero-diagonal constraint on $W$ in recommendation settings. Consequently, we chose not to explicitly discuss them in this paper, focusing instead on the more effective zero-diagonal constraint, which better aligns with the practical demands of recommendation tasks.

**Empirical Implication and Potential Applications of PAC-Bayes Bound for Recommendation Setting:** In implicit recommendation settings, the performance of recommendation models is typically evaluated using top-$k$ metrics such as Recall@$k$ or NDCG@$k$ during offline evaluation. However, optimizing these metrics directly is challenging due to their non-differentiable nature. As a result, recommendation models often rely on surrogate loss functions – for example, linear recommendation models commonly minimize the sum of squared element-wise errors. This reliance creates a potential mismatch, as the loss function optimized during training does not directly align with the metrics used for evaluation.

While PAC-Bayes bounds are derived for the surrogate loss (e.g., sum of squared errors), they can be recast to indirectly relate to evaluation metrics by decomposing the bound into two components: (1) a generalization bound on the surrogate loss, and (2) the empirical correlation between the surrogate loss and top-$k$ metrics. This decomposition provides a theoretical framework to quantify the mismatch and understand how improving generalization on the surrogate loss translates to better performance on top-$k$ metrics.

Additionally, since recommendation models depend on surrogate loss functions, ensuring that these functions generalize well to unseen data is critical. PAC-Bayes bounds offer guarantees on the generalization of surrogate losses, which is a necessary condition for achieving strong downstream performance. Thus, while surrogate losses do not directly align with top-$k$ metrics like Recall@$k$ or NDCG@$k$, demonstrating low generalization error on the surrogate loss provides a strong theoretical foundation for the model's ability to perform well on these evaluation metrics.

For instance, if a model performs poorly on the top-$k$ metrics, PAC-Bayes bounds can help identify whether the poor results are likely due to model uncertainty (indicated by a large bound) or other factors: 1) A large PAC-Bayes bound could indicate high model uncertainty, suggesting insufficient training or data sparsity. 2) A small PAC-Bayes bound coupled with poor performance might point to issues like suboptimal surrogate metrics, data distribution shifts, or model design.

**Quantifying the Gap between model loss and top $k$ metrics:** Finally, we would like to point that there is lack of formal analytical framework that links various loss functions to top-$k$ recommendation metrics in implicit settings. Establishing such a connection would bridge the divide between training objectives and evaluation metrics, potentially enabling the development of more effective recommendation models. We argue that addressing this challenge is an important open problem for both the recommendation systems and machine learning communities.

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

## A  PROOFS OF THE THEOREMS

*Proof of Theorem 1*:

Given $W$, let $(x, y) \sim \mathcal{D}$, and denote $v = y - Wx$, then $v \sim \mathcal{N}(\mu_W, \Sigma_W)$. Suppose there exists $Q \in \mathbb{R}^{p \times p}$ such that $\Sigma_W = QQ^T$. Such $Q$ exists since we can take $Q = \Sigma_W^{1/2} = S^T \Lambda^{1/2} S$, but we do not assume it to be unique. Let $\epsilon \sim \mathcal{N}(0, I)$, then we can write $v = Q\epsilon + \mu_W$. Thus,

$$
\begin{aligned}
R^{\text{true}}(W) = \mathbb{E}_{(x,y)\sim\mathcal{D}}\left[\|y - Wx\|_F^2\right] &= \mathbb{E}_\epsilon\left[\|Q\epsilon + \mu_W\|_F^2\right] = \mathbb{E}_\epsilon\left[(Q\epsilon + \mu_W)^T(Q\epsilon + \mu_W)\right] \\
&= \mathbb{E}_\epsilon[\epsilon^T Q^T Q\epsilon + \mu_W^T Q\epsilon + \epsilon^T Q^T \mu_W + \mu_W^T \mu_W] = \text{tr}(Q^T Q) + \mu_W^T \mu_W \\
&= \text{tr}(QQ^T) + \mu_W^T \mu_W = \text{tr}(\Sigma_W) + \mu_W^T \mu_W
\end{aligned}
\tag{10}
$$

Also, we can express the random variable $\|v\|_F^2$ in quadratic form (Representation 3.1a.1, (Mathai & Provost, 1992)):

$$
\begin{aligned}
\|v\|_F^2 = v^T v &= (Q\epsilon + \mu_W)^T(Q\epsilon + \mu_W) \\
&= (Q\epsilon + \mu_W)^T \Sigma_W^{-1/2} \Sigma_W \Sigma_W^{-1/2}(Q\epsilon + \mu_W) \\
&= (\Sigma_W^{-1/2}Q\epsilon + \Sigma_W^{-1/2}\mu_W)^T \Sigma_W (\Sigma_W^{-1/2}Q\epsilon + \Sigma_W^{-1/2}\mu_W) \\
&= (\Sigma_W^{-1/2}Q\epsilon + \Sigma_W^{-1/2}\mu_W)^T S^T \Lambda S (\Sigma_W^{-1/2}Q\epsilon + \Sigma_W^{-1/2}\mu_W) \\
&= (S\Sigma_W^{-1/2}Q\epsilon + S\Sigma_W^{-1/2}\mu_W)^T \Lambda (S\Sigma_W^{-1/2}Q\epsilon + S\Sigma_W^{-1/2}\mu_W)
\end{aligned}
$$

Denote $\epsilon' = S\Sigma_W^{-1/2}Q\epsilon$, then $\epsilon' \sim \mathcal{N}(0, I)$. This is because $\mathbb{E}[\epsilon'] = S\Sigma_W^{-1/2}Q\mathbb{E}[\epsilon] = 0$ and

$$
\text{Cov}[\epsilon'] = \mathbb{E}[\epsilon'\epsilon'^T] = S\Sigma_W^{-1/2}Q\mathbb{E}[\epsilon\epsilon^T]Q^T\Sigma_W^{-1/2}S^T = I
$$

As $b = S\Sigma_W^{-1/2}\mu_W$, we can write

$$
\|v\|_F^2 = (\epsilon' + b)^T \Lambda(\epsilon' + b) = \sum_{i=1}^p \eta_i(\epsilon'_i + b_i)^2
$$

Hence each $\epsilon'_i + b_i$ is independently from $\mathcal{N}(b_i, 1)$, and $(\epsilon'_i + b_i)^2$ is independently from the non-central chi-squared distribution of noncentrality parameter $b_i^2$ and with degree 1 of freedom. Thus the MGF of $(\epsilon'_i + b_i)^2$ is

$$
M_{(\epsilon'_i + b_i)^2}(t) = \mathbb{E}_{(\epsilon'_i + b_i)^2}[e^{t(\epsilon'_i + b_i)^2}] = \frac{\exp\left(\frac{b_i^2 t}{1-2t}\right)}{(1-2t)^{1/2}}
$$

Let $v_j = y_j - Wx_j$ such that $v_1, v_2, ..., v_m$ are i.i.d. from $\mathcal{N}(\mu_W, \Sigma_W)$, then

$$
R^{\text{emp}}(W) = \frac{1}{m}\sum_{j=1}^m \|y_j - Wx_j\|_F^2 = \frac{1}{m}\sum_{j=1}^m \|v_j\|_F^2
$$

Hence the MGF of $R^{\text{emp}}(W)$ is

$$
\begin{aligned}
M_{R^{\text{emp}}(W)}(t) &= \mathbb{E}_{S\sim\mathcal{D}^m}\left[e^{tR^{\text{emp}}(W)}\right] = \mathbb{E}_{S\sim\mathcal{D}^m}\left[\exp\left(\frac{t}{m}\sum_{j=1}^m \|v_j\|_F^2\right)\right] \\
&= \left(\mathbb{E}_{S\sim\mathcal{D}^m}\left[\exp\left(\frac{t}{m}\|v\|_F^2\right)\right]\right)^m = \left(\mathbb{E}_{S\sim\mathcal{D}^m}\left[\exp\left(\frac{t}{m}\sum_{i=1}^p \eta_i(\epsilon'_i + b_i)^2\right)\right]\right)^m \\
&= \left(\prod_{i=1}^p \mathbb{E}_{(\epsilon'_i+b_i)^2}\left[\exp\left(\frac{t\eta_i}{m}(\epsilon'_i + b_i)^2\right)\right]\right)^m = \left(\prod_{i=1}^p \frac{\exp\left(\frac{tb_i^2\eta_i}{m-2t\eta_i}\right)}{(1-2t\eta_i/m)^{1/2}}\right)^m \\
&= \frac{\exp\left(\sum_{i=1}^p \frac{tmb_i^2\eta_i}{m-2t\eta_i}\right)}{\prod_{i=1}^p (1-2t\eta_i/m)^{m/2}}
\end{aligned}
\tag{11}
$$

By Eq (10) and Eq (11), we can expand $\Psi_{\pi,\mathcal{D}}(\lambda, m)$ as

$$\Psi_{\pi,\mathcal{D}}(\lambda, m) = \ln \mathbb{E}_{W\sim\pi}\mathbb{E}_{S\sim\mathcal{D}^m}[e^{\lambda(R^{\text{true}}(W) - R^{\text{emp}}(W))}]$$

$$= \ln \mathbb{E}_{W\sim\pi}\left[e^{\lambda R^{\text{true}}(W)}\mathbb{E}_{S\sim\mathcal{D}^m}[e^{-\lambda R^{\text{emp}}(W)}]\right]$$

$$= \ln \mathbb{E}_{W\sim\pi}\left[\exp\left(\lambda\left(\text{tr}(\Sigma_W) + \mu_W^T\mu_W\right)\right) \frac{\exp\left(\sum_{i=1}^p \frac{-\lambda mb_i^2\eta_i}{m+2\lambda\eta_i}\right)}{\prod_{i=1}^p (1+2\lambda\eta_i/m)^{m/2}}\right] \quad (12)$$

Use the inequality that for any $x > 0$ and $k > 0$, $e^{\frac{xk}{x+k}} < (\frac{x}{k}+1)^k$ [1], and the fact $\text{tr}(\Sigma_W) = \sum_{i=1}^p \eta_i$, we have

$$\ln \mathbb{E}_{W\sim\pi}\left[\exp\left(\lambda\left(\text{tr}(\Sigma_W) + \mu_W^T\mu_W\right)\right) \frac{\exp\left(\sum_{i=1}^p \frac{-\lambda mb_i^2\eta_i}{m+2\lambda\eta_i}\right)}{\prod_{i=1}^p (1+2\lambda\eta_i/m)^{m/2}}\right]$$

$$\leq \ln \mathbb{E}_{W\sim\pi}\left[\exp\left(\lambda\left(\text{tr}(\Sigma_W) + \mu_W^T\mu_W\right)\right) \frac{\exp\left(\sum_{i=1}^p \frac{-\lambda mb_i^2\eta_i}{m+2\lambda\eta_i}\right)}{\prod_{i=1}^p \exp\left(\frac{m\lambda\eta_i}{m+2\lambda\eta_i}\right)}\right]$$

$$= \ln \mathbb{E}_{W\sim\pi}\exp\left(\lambda\mu_W^T\mu_W + \sum_{i=1}^p \lambda(\eta_i - \frac{mb_i^2\eta_i}{m+2\lambda\eta_i}) - \sum_{i=1}^p \frac{m\lambda\eta_i}{m+2\lambda\eta_i}\right)$$

$$= \ln \mathbb{E}_{W\sim\pi}\exp\left(\lambda\mu_W^T\mu_W + \sum_{i=1}^p \frac{2\lambda^2\eta_i^2 - \lambda mb_i^2\eta_i}{m+2\lambda\eta_i}\right)$$

$$\leq \ln \mathbb{E}_{W\sim\pi}\exp\left(\lambda(\mu_W^T\mu_W - \sum_{i=1}^p b_i^2\eta_i) + \frac{2\lambda^2(\sum_{i=1}^p \eta_i^2)}{m}\right) = \ln \mathbb{E}_{W\sim\pi}\exp\left(\frac{2\lambda^2(\sum_{i=1}^p \eta_i^2)}{m}\right)$$

The last equality above is because

$$\sum_{i=1}^p b_i^2\eta_i = b^T\Lambda b = \mu_W^T\Sigma_W^{-1/2}S^T\Lambda S\Sigma_W^{-1/2}\mu_W = \mu_W^T\mu_W$$

Since

$$\sum_{i=1}^p \eta_i^2 = \text{tr}(S^T\Lambda^2 S) = \text{tr}(\Sigma_W^2) = \text{tr}(\Sigma_W\Sigma_W^T) = \|\Sigma_W\|_F^2$$

we have

$$\ln \mathbb{E}_{W\sim\pi}\exp\left(\frac{2\lambda^2(\sum_{i=1}^p \eta_i^2)}{m}\right) = \ln \mathbb{E}_{W\sim\pi}\exp\left(\frac{2\lambda^2\|\Sigma_W\|_F^2}{m}\right)$$

$\square$

*Proof of Theorem 2*:

By Eq (12), we let $\{f_m\}_{m\in\mathbb{N}}$ be a sequence of functions where

$$f_m(W) = \exp\left(\lambda\left(\text{tr}(\Sigma_W) + \mu_W^T\mu_W\right)\right) \frac{\exp\left(\sum_{i=1}^p \frac{-\lambda mb_i^2\eta_i}{m+2\lambda\eta_i}\right)}{\prod_{i=1}^p (1+2\lambda\eta_i/m)^{m/2}}$$

for $m > 0$, and

$$f_0(W) = \exp\left(\lambda\left(\text{tr}(\Sigma_W) + \mu_W^T\mu_W\right)\right)$$

Note that each $f_i$ is a non-negative function.

---

[1]Since $\frac{x}{x+1} < \ln(x+1)$ for any $x > -1$, replacing $x$ with $\frac{x}{k}$, and taking exponential on both sides, we get $e^{\frac{xk}{x+k}} < (\frac{x}{k}+1)^k$.

Now we prove the following three conditions:

(1) $f_m(W) \leq f_0(W)$ for any $m$ and $W$.

Since $\lambda > 0$ and $\eta_i > 0$ for all $i$, we have $f_0(W) \geq f_1(W) \geq f_2(W)...$ for any $W$. This is because, when $W$ is fixed, the numerator $\exp\left(\sum_{i=1}^{p} \frac{-\lambda m b_i^2 \eta_i}{m+2\lambda\eta_i}\right)$ is monotonically decreasing with $m$ for $m \geq 0$, the denominator $\prod_{i=1}^{p}(1 + 2\lambda\eta_i/m)^{m/2}$ is monotonically increasing with $m$ for $m > 0$, and $(1 + 2\lambda\eta_i/m)^{m/2} \geq 1$ for any $m > 0$.

(2) $f_m \to 1$ pointwisely as $m \to \infty$.

For any $W$,

$$\lim_{m\to\infty} f_m(W) = \exp\left(\lambda\left(\text{tr}(\Sigma_W) + \mu_W^T\mu_W\right)\right) \lim_{m\to\infty} \frac{\exp\left(\sum_{i=1}^{p} \frac{-\lambda m b_i^2 \eta_i}{m+2\lambda\eta_i}\right)}{\prod_{i=1}^{p}(1 + 2\lambda\eta_i/m)^{m/2}}$$

$$= \exp\left(\lambda\left(\text{tr}(\Sigma_W) + \mu_W^T\mu_W\right)\right) \frac{\exp\left(\sum_{i=1}^{p} \lim_{m\to\infty} \frac{-\lambda m b_i^2 \eta_i}{m+2\lambda\eta_i}\right)}{\prod_{i=1}^{p} \lim_{m\to\infty}(1 + 2\lambda\eta_i/m)^{m/2}}$$

$$= \exp\left(\lambda\left(\text{tr}(\Sigma_W) + \mu_W^T\mu_W\right)\right) \frac{\exp\left(\sum_{i=1}^{p} -\lambda b_i^2 \eta_i\right)}{\prod_{i=1}^{p}\exp\left(\lambda\eta_i\right)} = 1$$

The last inequality uses the facts that $\sum_{i=1}^{p} b_i^2\eta_i = \mu_W^T\mu_W$ and $\sum_{i=1}^{p} \eta_i = \text{tr}(\Sigma_W)$.

(3) $\mathbb{E}[f_0] < \infty$.

$$\mathbb{E}[f_0] = \mathbb{E}\exp\left(\lambda\left(\text{tr}(\Sigma_W) + \mu_W^T\mu_W\right)\right)$$

$$= \mathbb{E}\exp\left(\lambda\left[\text{tr}((W^* - W)\Sigma_x(W^* - W)^T + \Sigma_e) + \|(W^* - W)\mu_x\|_F^2\right]\right)$$

$$= \mathbb{E}\exp\left(\lambda\left[\sum_{i=1}^{p}(W^* - W)_{i*}\Sigma_x(W^* - W)_{i*}^T + \text{tr}(\Sigma_e) + \sum_{i=1}^{p}(W^* - W)_{i*}\mu_x\mu_x^T(W^* - W)_{i*}^T\right]\right)$$

$$= \mathbb{E}\exp\left(\lambda\left[\sum_{i=1}^{p}(W^* - W)_{i*}\left[\Sigma_x + \mu_x\mu_x^T\right](W^* - W)_{i*}^T + \text{tr}(\Sigma_e)\right]\right)$$

$$= \mathbb{E}\exp\left(\lambda\left[\left\|\left(\Sigma_x + \mu_x\mu_x^T\right)^{1/2}(W^* - W)\right\|_F^2 + \text{tr}(\Sigma_e)\right]\right)$$

$$= \exp\left(\lambda\text{tr}(\Sigma_e)\right)\mathbb{E}\exp\left(\lambda\left[\left\|\left(\Sigma_x + \mu_x\mu_x^T\right)^{1/2}(W^* - W)\right\|_F^2\right]\right) < \infty$$

The last inequality holds because $\mathbb{E}\exp\left(\lambda\left[\left\|\left(\Sigma_x + \mu_x\mu_x^T\right)^{1/2}(W^* - W)\right\|_F^2\right]\right) < \infty$ is our assumption and $\exp\left(\lambda\text{tr}(\Sigma_e)\right)$ is a constant.

Denote $E = \mathbb{R}^{p\times p}$ such that $W \in E$. Since $W \sim \pi$, we consider $\pi$ as a probability measure $\mu$ on $E$ with $\mu(E) = 1$. Then we can express $\mathbb{E}[f_m]$ as a Lebesgue integral:

$$\mathbb{E}[f_m] = \int_E f_m d\mu$$

Also, condition (3) can be written as $\int_E f_0 d\mu < \infty$. Since the conditions (1), (2) and (3) hold, by the Dominated Convergence Theorem (Theorem 11.32, (Rudin, 1976) [2]), we have

$$\lim_{m\to\infty} \int_E f_m d\mu = \int_E \lim_{m\to\infty} f_m d\mu = \int_E 1 d\mu = 1$$

---

[2]Another version of the theorem is Theorem 5.3.3, (Resnick, 1998). We use Rudin's version since it makes the proof easier to understand.

Or equivalently,

$$\lim_{m\to\infty} \mathbb{E}\left[f_m\right] = \mathbb{E}\left[\lim_{m\to\infty} f_m\right] = \mathbb{E}[1] = 1$$

Since ln is continuous on $(0, \infty)$, we can interchange lim and ln. Therefore,

$$\lim_{m\to\infty} \Psi_{\pi,\mathcal{D}}(\lambda, m) \le \lim_{m\to\infty} \ln \mathbb{E}[f_m] = \ln \lim_{m\to\infty} \mathbb{E}[f_m] = \ln 1 = 0$$

$\square$

*Proof of Lemma 1*:

Let $X \sim \mathcal{N}(\mu, \sigma^2)$, then for any $t > 0$,

$$\mathbb{E}_X[tY_k] = \int \exp\left(t\sum_{i=0}^{k} a_i x^i\right) \frac{1}{\sqrt{2\pi}\sigma} \exp\left(-\frac{(x-\mu)^2}{2\sigma^2}\right) dx$$

$$= \frac{1}{\sqrt{2\pi}\sigma} \int \exp\left(t\sum_{i=0}^{k} a_i x^i - \frac{(x-\mu)^2}{2\sigma^2}\right) dx \qquad (13)$$

Since $k \ge 3$ and $a_k > 0$, $t\sum_{i=0}^{k} a_i x^i - \frac{(x-\mu)^2}{2\sigma^2}$ is a polynomial of $x$ with degree $\ge 3$, with leading coefficient being positive, thus

$$\lim_{x\to\infty} \exp\left(t\sum_{i=0}^{k} a_i x^i - \frac{(x-\mu)^2}{2\sigma^2}\right) = \infty$$

And the integral in Eq (13) is infinity.

$\square$

*Proof of Theorem 3*:

(1) Denote $V = \mathbb{E}_{W\sim\rho}[(I-W)(I-W)^T]$, then

$$\mathbb{E}_{W\sim\rho}[R^{\text{emp}}(W)] = \frac{1}{m}\mathbb{E}_{W\sim\rho}[\|R - RW\|_F^2] = \frac{1}{m}\sum_{i=1}^{m}\mathbb{E}_{W\sim\rho}[\|R_i - R_i W\|_F^2]$$

$$= \frac{1}{m}\sum_{i=1}^{m} R_i \mathbb{E}_{W\sim\rho}[(I-W)(I-W)^T]R_i^T = \frac{1}{m}\sum_{i=1}^{m} R_i V R_i^T$$

$V$ is a function of $\mathcal{U}$ and $\mathcal{S}$, i.e.,

$$V = \mathbb{E}_{W\sim\rho}[(I-W)(I-W)^T] = I - \mathbb{E}_{W\sim\rho}[W] - \mathbb{E}_{W\sim\rho}[W^T] + \mathbb{E}_{W\sim\rho}[WW^T]$$

$$= I - \mathcal{U} - \mathcal{U}^T + \left[\mathcal{U}\mathcal{U}^T + \text{diag}\left(\sum_{j=1}^{n}\mathcal{S}_{1j}, \sum_{j=1}^{n}\mathcal{S}_{2j}, ..., \sum_{j=1}^{n}\mathcal{S}_{nj}\right)\right]$$

$D(\rho\,||\,\pi)$ can also be written as a function of $\mathcal{U}$ and $\mathcal{S}$ by

$$D(\rho\,||\,\pi) = \frac{1}{2}\left[n^2(2\ln\sigma - 1) - \sum_{i=1}^{n}\sum_{j=1}^{n}(\ln\mathcal{S}_{ij} - \frac{\mathcal{S}_{ij}}{\sigma^2}) + \frac{\|\mathcal{U} - \mathcal{U}_0\|_F^2}{\sigma^2}\right]$$

Denote $f(\mathcal{U}, \mathcal{S}|\mathcal{U}_0, \sigma, \lambda) = \frac{1}{m}\sum_{i=1}^{m} R_i V R_i^T + \frac{1}{\lambda}D(\rho\,||\,\pi)$, our optimization problem becomes

$$\min_{\mathcal{U},\mathcal{S}} f(\mathcal{U}, \mathcal{S}|\mathcal{U}_0, \sigma, \lambda) \qquad (14)$$

The optimal $\mathcal{U}$ and $\mathcal{S}$ has closed-form solution, which can be obtained by solving $\frac{\partial}{\partial\mathcal{U}}f(\mathcal{U}, \mathcal{S}|\mathcal{U}_0, \sigma, \lambda) = 0$ and $\frac{\partial}{\partial\mathcal{S}}f(\mathcal{U}, \mathcal{S}|\mathcal{U}_0, \sigma, \lambda) = 0$.

First we show the partial derivatives of the $\frac{1}{\lambda}D(\rho\,||\,\pi)$ term:

$$\frac{\partial}{\partial \mathcal{U}_{ij}}\frac{1}{\lambda}D(\rho\,||\,\pi) = \frac{(\mathcal{U}_{ij}-(\mathcal{U}_0)_{ij})}{\lambda\sigma^2}\,, \quad \frac{\partial}{\partial \mathcal{S}_{ij}}\frac{1}{\lambda}D(\rho\,||\,\pi) = -\frac{1}{2\lambda}(\frac{1}{\mathcal{S}_{ij}}-\frac{1}{\sigma^2})$$

Then we discuss the partial derivatives of the $\frac{1}{m}\sum_{i=1}^m R_i V R_i^T$ term. Given $i$, for any $j$,

$$\frac{\partial}{\partial \mathcal{S}_{ij}}\frac{1}{m}\sum_{l=1}^m R_l V R_l^T = \frac{\partial}{\partial \mathcal{S}_{ij}}\frac{1}{m}\sum_{l=1}^m R_l \mathrm{diag}\left(\sum_{k=1}^n \mathcal{S}_{1k}, \sum_{k=1}^n \mathcal{S}_{2k}, ..., \sum_{k=1}^n \mathcal{S}_{nk}\right)R_l^T$$

$$= \frac{\partial}{\partial \mathcal{S}_{ij}}\frac{1}{m}\sum_{l=1}^m R_{li}\mathcal{S}_{ij}R_{li} = \frac{1}{m}\sum_{l=1}^m R_{li}^2 = \frac{1}{m}R_{*i}^T R_{*i}$$

Besides,

$$\frac{\partial}{\partial \mathcal{U}_{ij}}\frac{1}{m}\sum_{l=1}^m R_l V R_l^T = \frac{\partial}{\partial \mathcal{U}_{ij}}\frac{1}{m}\sum_{l=1}^m R_l(-\mathcal{U}-\mathcal{U}^T+\mathcal{U}\mathcal{U}^T)R_l^T$$

Since

$$\frac{\partial}{\partial \mathcal{U}_{ij}}\frac{1}{m}\sum_{l=1}^m R_l \mathcal{U} R_l^T = \frac{\partial}{\partial \mathcal{U}_{ij}}\frac{1}{m}\sum_{l=1}^m R_{li}\mathcal{U}_{ij}R_{lj} = \frac{1}{m}R_{*i}^T R_{*j}$$

$$\frac{\partial}{\partial \mathcal{U}_{ij}}\frac{1}{m}\sum_{l=1}^m R_l \mathcal{U}\mathcal{U}^T R_l^T = \frac{\partial}{\partial \mathcal{U}_{ij}}\frac{1}{m}\sum_{l=1}^m \sum_{k=1}^n (R_l \mathcal{U}_{*k})^2 = \frac{\partial}{\partial \mathcal{U}_{ij}}\frac{1}{m}\sum_{l=1}^m (R_l \mathcal{U}_{*j})^2 = \frac{2}{m}\sum_{l=1}^m R_{li}(R_l \mathcal{U}_{*j})$$

$$= \frac{2}{m}\sum_{l=1}^m (R_{li}R_l)\mathcal{U}_{*j} = \frac{2}{m}R_{*i}^T R\mathcal{U}_{*j}$$

Therefore,

$$\frac{\partial}{\partial \mathcal{U}_{ij}}\frac{1}{m}\sum_{l=1}^m R_l V R_l^T = -\frac{1}{m}R_{*i}^T R_{*j} - \frac{1}{m}R_{*j}^T R_{*i} + \frac{2}{m}R_{*i}^T R\mathcal{U}_{*j} = \frac{2}{m}\left(-R_{*i}^T R_{*j} + R_{*i}^T R\mathcal{U}_{*j}\right)$$

Wrap up the above results, we get

$$\frac{\partial}{\partial \mathcal{S}_{ij}}f(\mathcal{U},\mathcal{S}|\mathcal{U}_0,\sigma,\lambda) = \frac{1}{m}R_{*i}^T R_{*i} - \frac{1}{2\lambda}(\frac{1}{\mathcal{S}_{ij}}-\frac{1}{\sigma^2}) \tag{15}$$

$$\frac{\partial}{\partial \mathcal{U}_{ij}}f(\mathcal{U},\mathcal{S}|\mathcal{U}_0,\sigma,\lambda) = \frac{2}{m}\left(-R_{*i}^T R_{*j} + R_{*i}^T R\mathcal{U}_{*j}\right) + \frac{(\mathcal{U}_{ij}-(\mathcal{U}_0)_{ij})}{\lambda\sigma^2} \tag{16}$$

Therefore, the solution of $\frac{\partial}{\partial \mathcal{S}}f(\mathcal{U},\mathcal{S}|\mathcal{U}_0,\sigma,\lambda) = 0$ is that, for any $i = 1,2,...,n$,

$$\mathcal{S}_{ij} = \frac{1}{\frac{2\lambda}{m}R_{*i}^T R_{*i} + \frac{1}{\sigma^2}} \quad \text{for } j = 1,2,...,n \tag{17}$$

By Eq (16) we have

$$\frac{\partial}{\partial \mathcal{U}}f(\mathcal{U},\mathcal{S}|\mathcal{U}_0,\sigma,\lambda) = \left[\frac{2}{m}(-R^T R + R^T R\mathcal{U}) + \frac{1}{\lambda\sigma^2}(\mathcal{U}-\mathcal{U}^0)\right]^T \tag{18}$$

Thus the solution of $\frac{\partial}{\partial \mathcal{U}}f(\mathcal{U},\mathcal{S}|\mathcal{U}_0,\sigma,\lambda) = 0$ is

$$\mathcal{U} = \left(\frac{1}{m}R^T R + \frac{1}{2\lambda\sigma^2}I\right)^{-1}\left(\frac{1}{m}R^T R + \frac{1}{2\lambda\sigma^2}\mathcal{U}_0\right) \tag{19}$$

Now we show that $f(\mathcal{U},\mathcal{S}|\mathcal{U}_0,\sigma,\lambda)$ is a convex function, thus the solutions of $\mathcal{S}$ in Eq (17) and $\mathcal{U}$ in Eq (19) are the global minimizer of Eq (14). Denote $\nu \in \mathbb{R}^{2n^2}$ where for $i = 1,2,...,n$ and $j =$

$1, 2, ..., n$, $\nu_{(i-1)n+j} = \mathcal{U}_{ij}$ and $\nu_{n^2+(i-1)n+j} = \mathcal{S}_{ij}$. Let $H_f \in \mathbb{R}^{2n^2 \times 2n^2}$ be the Hessian matrix where $(H_f)_{ij} = \frac{\partial^2 f}{\partial \nu_i \partial \nu_j}$. Then we can write $H_f = \begin{bmatrix} A & 0 \\ 0 & B \end{bmatrix}$ where $A = (R^T R) \otimes I_n + \frac{1}{\lambda \sigma^2} I_{n^2}$ and $B$ is a $n^2 \times n^2$ diagonal matrix with $B_{(i-1)n+j,(i-1)n+j} = \frac{1}{2\lambda(\mathcal{S}_{ij})^2}$. Here $\otimes$ means Kronecker product.

The Kronecker product has a property that, let $\{\lambda_i | i = 1, ..., m\}$ be the eigenvalues of $A \in \mathbb{R}^{m \times m}$ and $\{\mu_j | j = 1, ..., n\}$ be the eigenvalues of $B \in \mathbb{R}^{n \times n}$, then $\{\lambda_i \mu_j | i = 1, ..., m, j = 1, ..., n\}$ are the eigenvalues of $A \otimes B$ (Theorem 4.2.12, (Horn & Johnson, 1991)). Since $R^T R$ is positive semi-definite and $I_n$ is positive definite, $(R^T R) \otimes I_n$ is positive semi-definite. Thus $A$ is positive definite. Since all elements of $\mathcal{S}$ is positive, $B$ is positive definite. Therefore, $H_f$ is a positive definite matrix for any $\mathcal{U}$ and $\mathcal{S}$, which means $f(\mathcal{U}, \mathcal{S}|\mathcal{U}_0, \sigma, \lambda)$ is a convex function. Thus, the solutions of $\mathcal{S}$ in Eq (17) and $\mathcal{U}$ in Eq (19) give the global minimum.

(2) Since applying $\text{diag}(W) = 0$ to $\rho$ and $\pi$ is equivalent to set $\text{diag}(\mathcal{U}) = 0$, $\text{diag}(\mathcal{S}) = 0$, $\text{diag}(\mathcal{U}_0) = 0$, and $\text{diag}(\sigma^2 J) = 0$, the $D(\rho \,\|\, \pi)$ term in $f(\mathcal{U}, \mathcal{S}|\mathcal{U}_0, \sigma, \lambda)$ is changed to $D(\rho \,\|\, \pi) = \frac{1}{2} \left[ (n^2 - n)(2\ln \sigma - 1) - \sum_{i=1}^{n} \sum_{j=1, j \neq i}^{n} (\ln \mathcal{S}_{ij} - \frac{\mathcal{S}_{ij}}{\sigma^2}) + \frac{\|\mathcal{U} - \mathcal{U}_0\|_F^2}{\sigma^2} \right]$. In this case, Eq (15) holds only for $i \neq j$.

We let $\mathcal{S}_{11}, \mathcal{S}_{22}, ..., \mathcal{S}_{nn}$ be zero constants in $f(\mathcal{U}, \mathcal{S}|\mathcal{U}_0, \sigma, \lambda)$, and consider only the off-diagonal elements of $\mathcal{S}$ to be variables. Then we construct the Lagrangian function as

$$L(\mathcal{U}, \mathcal{S}, x|\mathcal{U}_0, \sigma, \lambda) = f(\mathcal{U}, \mathcal{S}|\mathcal{U}_0, \sigma, \lambda) + x^T \text{diag}(\mathcal{U})$$

for some $x \in \mathbb{R}^n$, and solve

$$\frac{\partial L}{\partial x} = [\text{diag}(\mathcal{U})]^T = 0 \tag{20}$$

$$\frac{\partial L}{\partial \mathcal{U}} = \left[ \frac{\partial}{\partial \mathcal{U}} f(\mathcal{U}, \mathcal{S}|\mathcal{U}_0, \sigma, \lambda) + \text{Diag}(x) \right]^T = 0 \tag{21}$$

$$\frac{\partial L}{\partial \mathcal{S}_{ij}} = \frac{\partial}{\partial \mathcal{S}_{ij}} f(\mathcal{U}, \mathcal{S}|\mathcal{U}_0, \sigma, \lambda) = 0 \quad \text{for } i, j \in \{1, 2, ..., n\}, i \neq j \tag{22}$$

The optimal $\mathcal{S}$ is obtained by solving Eq (22) and set $S_{ii} = 0$ for all $i$. The solution of Eq (22) is Eq (17) with $i \neq j$. The optimal $\mathcal{U}$ is obtained by solving Eq (21) and Eq (20). By Eq (21),

$$\frac{2}{m}(-R^T R + R^T R \mathcal{U}) + \frac{1}{\lambda \sigma^2}(\mathcal{U} - \mathcal{U}^0) + \text{Diag}(x) = 0$$

$$\Longleftrightarrow \mathcal{U} = \left( \frac{1}{m} R^T R + \frac{1}{2\lambda\sigma^2} I \right)^{-1} \left( \frac{1}{m} R^T R + \frac{1}{2\lambda\sigma^2} \mathcal{U}_0 - \frac{1}{2} \text{Diag}(x) \right) \tag{23}$$

Then we solve $x$ to satisfy Eq (20),

$$\text{diag}(\mathcal{U}) = \text{diag}\left[ \left( \frac{1}{m} R^T R + \frac{1}{2\lambda\sigma^2} I \right)^{-1} \left( \frac{1}{m} R^T R + \frac{1}{2\lambda\sigma^2} \mathcal{U}_0 \right) \right] - \text{diag}\left[ \frac{1}{2} \left( \frac{1}{m} R^T R + \frac{1}{2\lambda\sigma^2} I \right)^{-1} \text{Diag}(x) \right]$$

$$= \text{diag}\left[ \left( \frac{1}{m} R^T R + \frac{1}{2\lambda\sigma^2} I \right)^{-1} \left( \frac{1}{m} R^T R + \frac{1}{2\lambda\sigma^2} \mathcal{U}_0 \right) \right] - \frac{1}{2} \text{diag}\left[ \left( \frac{1}{m} R^T R + \frac{1}{2\lambda\sigma^2} I \right)^{-1} \right] \odot x = 0$$

we get

$$x = 2 \cdot \text{diag}\left[ \left( \frac{1}{m} R^T R + \frac{1}{2\lambda\sigma^2} I \right)^{-1} \left( \frac{1}{m} R^T R + \frac{1}{2\lambda\sigma^2} \mathcal{U}_0 \right) \right] \oslash \text{diag}\left[ \left( \frac{1}{m} R^T R + \frac{1}{2\lambda\sigma^2} I \right)^{-1} \right]$$

To show the solution of Eq (20), Eq (21) and Eq (22) gives the global minimum of the problem Eq (14) under the constraint $\text{diag}(W) = 0$, we use the lemma that if the Hessian matrix $H_L$ where $(H_L)_{ij} = \frac{\partial^2 L}{\partial \nu_i \partial \nu_j}$ is positive definite for any $\mathcal{U}, \mathcal{S}, x$, then any solution of $\frac{\partial L}{\partial \mathcal{U}} = 0, \frac{\partial L}{\partial \mathcal{S}} = 0, \frac{\partial L}{\partial x} = 0$ will satisfy the second order sufficient conditions (Section 11.5, (Luenberger & Ye, 2008)), thus becomes a local minimizer.

It is easy to show that if we remove the dimensions corresponding to $\mathcal{S}_{11}, \mathcal{S}_{22}, ...\mathcal{S}_{nn}$ of $H_f$ and get $H_f' \in \mathbb{R}^{(2n^2-n)\times(2n^2-n)}$, then $H_f'$ will be equivalent to $H_L$. Thus $H_L$ is always positive definite. Since the solution of Eq (20), Eq (21) and Eq (22) is unique, it gives the global minimum.

$\square$

*Proof of Theorem 4*:

Let $P, Q \in \mathbb{R}^{n \times n}$ be two symmetric matrices, we write $P \succeq Q$ if $P - Q$ is positive semi-definite and $P \succ Q$ if $P - Q$ is positive definite.

Let $\eta_j$ be the $j$th largest eigenvalue of $A$ and $\eta_j^{(i)}$ be the $j$th largest eigenvalue of $A^{(i)}$. By Corollary 7.7.4 (c) of (Horn & Johnson, 2012), $P \succeq Q$ implies $\eta_j(P) \geq \eta_j(Q)$ for any $j$. Since $A - A^{(i)} = \sigma^2(\Sigma_r^{1/2})_{*i}(\Sigma_r^{1/2})_{*i}^T \succeq 0$ for any $i$, we have $\eta_j \geq \eta_j^{(i)}$ for any $i, j$.

Since $b^{(i)} = S^{(i)}(A^{(i)})^{-1/2}\mu^i$, we have

$$(b_j^{(i)})^2 \eta_j^{(i)} = \eta_j^{(i)}(\mu^i)^T(A^{(i)})^{-1/2}(S_{j*}^{(i)})^T S_{j*}^{(i)}(A^{(i)})^{-1/2}\mu^i$$
$$= \eta_j^{(i)}(\mu^i)^T(S^{(i)})^T(\Lambda^{(i)})^{-1/2}[S^{(i)}(S_{j*}^{(i)})^T][S_{j*}^{(i)}(S^{(i)})^T](\Lambda^{(i)})^{-1/2}(S^{(i)})\mu^i$$
$$= (\mu^i)^T(S_{j*}^{(i)})^T(S_{j*}^{(i)})\mu^i$$

Therefore,

$$\mathbb{E}_\pi\left[e^{\lambda R^{\text{true}}(W)}\right] = \prod_{i=1}^n \prod_{j=1}^n \frac{\exp\left(\frac{t(b_j^{(i)})^2\eta_j^{(i)}}{1-2\lambda\eta_j}\right)}{\left(1-2\lambda\eta_j^{(i)}\right)^{1/2}} = \prod_{i=1}^n \prod_{j=1}^n \frac{\exp\left(\frac{\lambda(\mu^i)^T(S_{j*}^{(i)})^T(S_{j*}^{(i)})\mu^i}{1-2\lambda\eta_j}\right)}{\left(1-2\lambda\eta_j^{(i)}\right)^{1/2}}$$

$$= \prod_{i=1}^n \frac{\exp\left(\lambda(\mu^i)^T\left(\sum_{j=1}^n \frac{(S_{j*}^{(i)})^T(S_{j*}^{(i)})}{1-2\lambda\eta_j}\right)\mu^i\right)}{\prod_{j=1}^n\left(1-2\lambda\eta_j^{(i)}\right)^{1/2}} = \prod_{i=1}^n \frac{\exp\left(\lambda(\mu^i)^T(S^{(i)})^T\bar{\Lambda}^{(i)}S^{(i)}\mu^i\right)}{\prod_{j=1}^n\left(1-2\lambda\eta_j^{(i)}\right)^{1/2}}$$

where $\bar{\Lambda}^{(i)} = \text{diag}\left(\frac{1}{1-2\lambda\eta_1^{(i)}}, \frac{1}{1-2\lambda\eta_2^{(i)}}, ..., \frac{1}{1-2\lambda\eta_n^{(i)}}\right)$.

Similarly we have

$$\mathbb{E}_{\pi'}\left[e^{\lambda R^{\text{true}}(W)}\right] = \prod_{i=1}^n \frac{\exp\left(\lambda(\mu^i)^T S^T\bar{\Lambda}S\mu^i\right)}{\prod_{j=1}^n(1-2\lambda\eta_j)^{1/2}}$$

where $\bar{\Lambda} = \text{diag}\left(\frac{1}{1-2\lambda\eta_1}, \frac{1}{1-2\lambda\eta_2}, ..., \frac{1}{1-2\lambda\eta_n}\right)$.

Now we show that $S^T\bar{\Lambda}S \succeq (S^{(i)})^T\bar{\Lambda}^{(i)}S^{(i)}$ for any $i$. By Corollary 7.7.4 (a) of (Horn & Johnson, 2012), if $P \succ 0$ and $Q \succ 0$, then $P \succeq Q$ if and only if $Q^{-1} \succeq P^{-1}$. Since we assume $0 < \lambda < \frac{1}{2\eta_1}$, we have $1 - 2\lambda\eta_j^{(i)} > 0$ and $1 - 2\lambda\eta_j > 0$ for any $i, j$, thus all diagonal elements of $\bar{\Lambda}^{(i)}$ and $\bar{\Lambda}$ are positive, implying that $(S^{(i)})^T\bar{\Lambda}^{(i)}S^{(i)} \succ 0$ and $S^T\bar{\Lambda}S \succ 0$.

Since $\left((S^{(i)})^T\bar{\Lambda}^{(i)}S^{(i)}\right)^{-1} = (S^{(i)})^T\left(I - 2\lambda\Lambda^{(i)}\right)S^{(i)} = I - 2\lambda A^{(i)}$ and $\left(S^T\bar{\Lambda}S\right)^{-1} = I - 2\lambda A$, we have

$$\left((S^{(i)})^T\bar{\Lambda}^{(i)}S^{(i)}\right)^{-1} \succeq \left(S^T\bar{\Lambda}S\right)^{-1} \iff I - 2\lambda A^{(i)} \succeq I - 2\lambda A \iff A \succeq A^{(i)}$$

Thus $S^T\bar{\Lambda}S \succeq (S^{(i)})^T\bar{\Lambda}^{(i)}S^{(i)}$, implying that $(\mu^i)^T S^T\bar{\Lambda}S\mu^i \geq (\mu^i)^T(S^{(i)})^T\bar{\Lambda}^{(i)}S^{(i)}\mu^i$ holds for any $\mu^i$. Therefore,

$$\mathbb{E}_\pi\left[e^{\lambda R^{\text{true}}(W)}\right] = \prod_{i=1}^n \frac{\exp\left(\lambda(\mu^i)^T(S^{(i)})^T\bar{\Lambda}^{(i)}S^{(i)}\mu^i\right)}{\prod_{j=1}^n\left(1-2\lambda\eta_j^{(i)}\right)^{1/2}} \leq \prod_{i=1}^n \frac{\exp\left(\lambda(\mu^i)^T S^T\bar{\Lambda}S\mu^i\right)}{\prod_{j=1}^n(1-2\lambda\eta_j)^{1/2}} = \mathbb{E}_{\pi'}\left[e^{\lambda R^{\text{true}}(W)}\right]$$

$\square$

## B   ALLOWING MULTIPLE TRAILS ON $\lambda$

Since we do not know the optimal value of $\lambda$, by the suggestions of (Alquier, 2021), we can choose a finite grid in $(0, +\infty)$ and search $\lambda$ in the grid. Let $L = \{\lambda_1, \lambda_2, ..., \lambda_l\}$ be the grid where each $\lambda_i > 0$ and $l = |L|$ is the cardinality of $L$.

$$P\left(\forall \lambda \in L, \quad \mathbb{E}_{W \sim \rho}[R^{\text{true}}(W)] < \mathbb{E}_{W \sim \rho}[R^{\text{emp}}(W)] + \frac{1}{\lambda}\left[D(\rho \| \pi) + \ln\frac{|L|}{\delta} + \Psi_{\pi, \mathcal{D}}(\lambda, m)\right]\right) \geq 1 - \delta$$

This is because

$$P\left(\forall \lambda \in L, \quad \mathbb{E}_{W \sim \rho}[R^{\text{true}}(W)] < \mathbb{E}_{W \sim \rho}[R^{\text{emp}}(W)] + \frac{1}{\lambda}\left[D(\rho \| \pi) + \ln\frac{|L|}{\delta} + \Psi_{\pi, \mathcal{D}}(\lambda, m)\right]\right)$$

$$= 1 - P\left(\exists \lambda \in L, \quad \mathbb{E}_{W \sim \rho}[R^{\text{true}}(W)] > \mathbb{E}_{W \sim \rho}[R^{\text{emp}}(W)] + \frac{1}{\lambda}\left[D(\rho \| \pi) + \ln\frac{|L|}{\delta} + \Psi_{\pi, \mathcal{D}}(\lambda, m)\right]\right)$$

$$= 1 - P\left(\bigcup_{i=1}^{|L|} \mathbb{E}_{W \sim \rho}[R^{\text{true}}(W)] > \mathbb{E}_{W \sim \rho}[R^{\text{emp}}(W)] + \frac{1}{\lambda_i}\left[D(\rho \| \pi) + \ln\frac{|L|}{\delta} + \Psi_{\pi, \mathcal{D}}(\lambda_i, m)\right]\right)$$

$$\geq 1 - \sum_{i=1}^{|L|} P\left(\mathbb{E}_{W \sim \rho}[R^{\text{true}}(W)] > \mathbb{E}_{W \sim \rho}[R^{\text{emp}}(W)] + \frac{1}{\lambda_i}\left[D(\rho \| \pi) + \ln\frac{|L|}{\delta} + \Psi_{\pi, \mathcal{D}}(\lambda_i, m)\right]\right)$$

$$\geq 1 - \sum_{i=1}^{|L|} \frac{\delta}{|L|} = 1 - \delta$$

## C   THE ERROR BETWEEN $\hat{\Sigma}_r$ AND $\Sigma_r$

We discuss how to measure the error between $(\hat{\Sigma}_r)_{ij}$ and $(\Sigma_r)_{ij}$ for any $i, j$. Suppose $\mathcal{D}$ is a bounded distribution such that for $r \sim \mathcal{D}$, $r_i \in [a, b]$ for any $i$. Let $c = \max\{|a|, |b|\}$, then each element in $R'^T_{i*}R'_{i*}$ is within the range $[0, c^2]$.

One way to measure the error is to use theoretical bounds based on concentration inequalities. For example, by Hoeffding's Inequality,

$$P\left(\left|(\hat{\Sigma}_r)_{ij} - (\Sigma_r)_{ij}\right| > t\right) \leq 2\exp\left(-\frac{2t^2 m'}{c^2}\right) \iff P\left(\left|(\hat{\Sigma}_r)_{ij} - (\Sigma_r)_{ij}\right| < c\sqrt{\frac{\ln(2/\delta)}{2m'}}\right) > 1 - \delta$$

where we let $\delta = 2\exp\left(-\frac{2t^2 m'}{c^2}\right)$. Such bounds are rigorous but tend to be vacuous. Further theoretical bounds based on matrix concentration inequalities can be found in (Tropp et al., 2015).

Another way is to use empirical bounds based on interval estimation. By the Popoviciu's inequality (Bhatia & Davis, 2000), the variance of each element in $R'^T_{i*}R'_{i*}$ is within the range $[0, c^2/4]$. Therefore, by central limit theorem, we have $\sqrt{m'}\left((\hat{\Sigma}_r)_{ij} - (\Sigma_r)_{ij}\right) \xrightarrow{d} \mathcal{N}(0, \sigma^2_{ij})$ for any $i, j$, where $\sigma^2_{ij} \leq c^2/4$. For large enough $m'$, a 99.7% confidence interval would be

$$P\left(\left|(\hat{\Sigma}_r)_{ij} - (\Sigma_r)_{ij}\right| < \frac{3c^2}{4m'}\right) = P\left((\hat{\Sigma}_r)_{ij} - \frac{3c^2}{4m'} < (\Sigma_r)_{ij} < (\hat{\Sigma}_r)_{ij} + \frac{3c^2}{4m'}\right)$$

$$\geq P\left((\hat{\Sigma}_r)_{ij} - \frac{3\sigma^2_{ij}}{4m'} < (\Sigma_r)_{ij} < (\hat{\Sigma}_r)_{ij} + \frac{3\sigma^2_{ij}}{4m'}\right) > 0.997$$

Note that this bound is not for theoretical use since it does not describe with how large $m'$ the bound will be satisfied. One commonly used rule of thumb is $m' > 30$.

Here we compare the two bounds: taking $\delta = 0.003$, $m' = 100000$ and $c = 5$, the first bound gives $c\sqrt{\frac{\ln(2/\delta)}{2m'}} \approx 0.0285$, while the second bound gives $\frac{3c^2}{4m'} \approx 0.00019$.

## D   RELATED WORKS

The earliest PAC-Bayes bound is proposed by (McAllester, 1998). (Alquier et al., 2016) proposed an oracle PAC-Bayes bound based under Hoeffding assumption. (Germain et al., 2016) applied

Alquier's bound to linear regression problem under Gaussian data and parameter distribution assumptions, but the bound does not converge for being independent of the number of samples. (Shalaeva et al., 2020) improved Germain's bound by proposing a bound related to the number of samples, and showed the bound converges as the number of samples increases. Most PAC-Bayes bounds are theoretical and difficult to calculate in practice, and some research is focused on making the bound more practical to compute. (Dziugaite & Roy, 2017) proposed a practical way to calculate Seeger's bound (Langford & Seeger, 2001) for neural networks, and showed the bound is nonvacuous on MNIST dataset, where the bound is around $10\times$ of the test error.

Recent years LAEs gains popularity in recommendation systems (particularly on implicit settings) due to their simplicity and effectiveness. (Steck, 2019) proposed the EASE model and showed it surpasses the performance of deep neural network models on recommendation datasets under Recall and NDCG metrics. Later (Steck, 2020) proposed EDLAE which introduces a mask to the target function to avoid the parameter matrix overfitting towards identity. (Vančura et al., 2022) proposed ELSA which constructs the LAE with an item-item similarity matrix $AA^T - I$ with zero diagonal.

Most LAE based recommender models constraints the diagonal of the weight matrix to zero. The zero diagonal constraint is closely related to the trace norm, which is considered as an effective tool for matrix completion. (Srebro & Salakhutdinov, 2010) applied the weighted traced norm in collaborative filtering. (Shamir & Shalev-Shwartz, 2014) proposed a sample complexity bound for the trace norm in matrix completion.

Another type of linear recommendation model is based on matrix factorization, which can be viewed as a form of low-rank matrix completion (Candes & Tao, 2009; Recht, 2011; Chen et al., 2014; Srebro & Shraibman, 2005; Foygel et al., 2011; Shamir & Shalev-Shwartz, 2011). Matrix factorization methods have been shown to be highly effective in explicit settings (Koren et al., 2009), where user preferences are explicitly expressed (e.g., ratings). However, they have been found to be less effective than LAE models in implicit settings (Cremonesi & Jannach, 2021; Jin et al., 2021), where interactions are inferred from user behavior (e.g., clicks or purchases).

Some studies have investigated the generalizability of the matrix factorization models. (Srebro et al., 2004) proposed a PAC bound based on covering number for collaborative filtering. Other generalization bounds include (Ledent et al., 2021) for inductive matrix completion and (Ledent & Alves, 2024) for deep non-linear matrix completion.

# E    SUPPLEMENTAL EXPERIMENT RESULTS

Table 3: Details of the terms in the PAC-Bayes bound in Table 2

| | Dataset | MovieLens 20M | Netflix | Yelp2018 | MSD |
|---|---|---|---|---|---|
| $\gamma = 50$ | $\lambda$ | 32 | 32 | 512 | 32 |
| | $\mathbb{E}_{W\sim\rho}[R^{\text{emp}}(W)]$ | 789.86 | 1412.10 | 30.41 | 1151.06 |
| | $D(\rho \,\|\, \pi)$ | 0.0888 | 0.1011 | 0.0036 | 0.7033 |
| | approx $\left(\Psi'_{\pi',\mathcal{D}}(\lambda)\right)$ | 28310.00 | 46649.43 | 15910.84 | 41133.95 |
| $\gamma = 100$ | $\lambda$ | 32 | 32 | 512 | 32 |
| | $\mathbb{E}_{W\sim\rho}[R^{\text{emp}}(W)]$ | 807.22 | 1412.92 | 29.77 | 1148.82 |
| | $D(\rho \,\|\, \pi)$ | 0.0889 | 0.1011 | 0.0037 | 0.7033 |
| | approx $\left(\Psi'_{\pi',\mathcal{D}}(\lambda)\right)$ | 28439.91 | 46626.98 | 16068.41 | 41116.51 |
| $\gamma = 200$ | $\lambda$ | 32 | 32 | 512 | 32 |
| | $\mathbb{E}_{W\sim\rho}[R^{\text{emp}}(W)]$ | 827.25 | 1414.95 | 28.90 | 1149.86 |
| | $D(\rho \,\|\, \pi)$ | 0.0889 | 0.1011 | 0.0037 | 0.7033 |
| | approx $\left(\Psi'_{\pi',\mathcal{D}}(\lambda)\right)$ | 28709.69 | 46618.04 | 16292.60 | 41131.17 |
| $\gamma = 400$ | $\lambda$ | 32 | 32 | 512 | 32 |
| | $\mathbb{E}_{W\sim\rho}[R^{\text{emp}}(W)]$ | 850.02 | 1419.18 | 27.86 | 1150.58 |
| | $D(\rho \,\|\, \pi)$ | 0.0891 | 0.1011 | 0.0039 | 0.7033 |
| | approx $\left(\Psi'_{\pi',\mathcal{D}}(\lambda)\right)$ | 29106.71 | 46652.54 | 16578.26 | 41213.43 |

