# OpenReview forum: "On PAC-Bayes Bounds for Linear Autoencoders"
_ICLR.cc/2025/Conference — Submitted to ICLR 2025_

### Official Review · Reviewer_Yb5B · 2024-10-29

**Soundness:** 3
**Presentation:** 3
**Contribution:** 2
**Rating:** 6
**Confidence:** 4

**Summary:**

This paper presents a PAC-Bayesian analysis of Linear Autoencoders (LAE), which have shown remarkable effectiveness in recommendation systems. The main contributions include:

* A thorough theoretical revision of convergence analysis from previous work (AAAI 2020), addressing and correcting significant mathematical inconsistencies in the derivations.
* Introduction of two novel theoretical bounds for LAE performance: Gaussian data distributions & data with Gaussian parameters. Both accompanied by computationally efficient implementation methods.
* Empirical validation on major recommendation datasets (MovieLens 20M, Netflix, Yelp2018, and MSD), demonstrating notably tight bounds that fall within twice the actual test error - a significant improvement over typical theoretical bounds in the field.

**Strengths:**

**Originality:**

*   Novel Application of PAC-Bayes Bounds: The paper's most original contribution lies in its application of PAC-Bayes bounds to analyze the generalizability of LAE models. While PAC-Bayes bounds have been used in other machine learning domains, their use in the specific context of LAE recommender systems is novel. The authors successfully bridge this gap by adapting existing PAC-Bayes frameworks, specifically Shalaeva's bound, to the challenges posed by LAEs. Specifically they had to extend  the analysis for single linear regression problems, to accommodate the multi-regression nature of LAEs. In LAEs, each row of the data matrix is treated as a separate regression problem, requiring a generalization of the bound to handle this specific structure. Here, I think it may exists similar approches in the bandit literature for the analysis of LinUCB like algorithms.


**Quality:**

*   Rigorous Theoretical Analysis: The paper exhibits a high level of quality through its meticulous theoretical derivations and proofs. The authors provide detailed steps for each proof, ensuring the mathematical soundness of their work. This rigorous approach strengthens the credibility of the proposed bounds and enhances the overall quality of the paper.
*   Development of a practical method for calculating the PAC-Bayes bound based on the bounded data and Gaussian parameter assumptions. This practical contribution bridges the gap between theory and practice, making the PAC-Bayes bound a useful tool for evaluating LAEs in real-world settings. The authors' adaptation of this method to the specific constraints of the EASE model.
*   Empirical Validation: The authors take a commendable step to empirically validate their theoretical findings by conducting experiments on four real-world datasets.  Their choice of datasets, including MovieLens 20M, Netflix, Yelp2018, and MSD, represents a diverse range of recommendation scenarios. The observed tightness of the bound, being within twice the test error is interesting (but the non vaccous claiming is an overclaim in my opinion)

**Clarity:**

*  Paper is well written and provide careful definition of the concepts and notation.
*  The authors honestly acknowledge the limitations of their work, particularly regarding the applicability of their bounds to more complex recommendation scenarios that use evaluation metrics such as Recall@k and NDCG@k.

**Significance:**

* **Identification and Resolution of Convergence Issues:** The paper demonstrates significance in its critical examination of Shalaeva's bound. In my point of view the errors made by Shalaeva's work are unacceptable (limit and integral inversion without any check and omitting a distribution when computing an E). They should at least lead to the withdrawal of the 2020 paper.
* Yet another linear analysis

**Weaknesses:**

1. The paper's positioning relative to previous work in recommender systems is inadequate and imprecise. For instance, citing Rendle 2022 for Matrix Factorization/ALS is an unusual choice, as Rendle is primarily known for his 2010 work on Factorization Machines. This suggests a need for more thorough engagement with the historical development of these methods.

2. The practical applicability of these findings to real-world recommender systems is unclear. While the experimental section successfully demonstrates the theoretical bounds, it falls short of the standards expected in recommender systems research, lacking comprehensive evaluation on metrics and scenarios that matter in practice.

3. The analysis primarily focuses on the non-regularized model, which has limited practical relevance as real-world systems invariably use regularization. Though the appendix addresses the regularized case, the analysis remains incomplete and doesn't provide meaningful insights into why this form of regularization is effective in practice.

**Questions:**

1. Since the initial error appears in a paper published at AAAI, why not contact the Program Committee from that edition? I believe AAAI should take responsibility for errors made during their review process.

2. I'm curious about how similar or different this is to the analysis of random projections and LinUCB. While we're working in a Bayesian framework, the mathematical tools used are quite similar, and some research has already bridged the gap between these approaches (such as the work in https://www.jmlr.org/papers/volume17/14-087/14-087.pdf).

3. I wonder if your bound could be helpful for optimistic exploration strategies when the user (row) is sampled from a uniform distribution and the item is selected by the algorithm. This might be challenging since independence is broken, and you'd likely need additional assumptions (such as rank-1 matrix only) - which, while commonly used in state-of-the-art approaches, may be overly simplistic.

4. In all EASE-based system implementations I'm aware of, the zero-diagonal constraint is used. While the paper studies this to some extent, can your analysis provide new insights beyond confirming it as a useful bias?

5. You mention producing 'non-vacuous' bounds. Could you elaborate on what constitutes such a bound? Specifically, if your definition implies practical usefulness, could you explain how these bounds could be used to improve recommender systems?

6. I suspect the relative tightness shown in the final table is due to M, which bounds the error on the test set. While getting a concentration-like inequality is expected, what additional insights does your analysis provide? From a practical perspective, M significantly reduces the bound. Could you discuss how M is computed (is it updated after each sample or calculated once using all data, including test data)? Also, a graph showing how the bound and test error vary with the number of observations would be more informative than a table that doesn't indicate how these results compare to the state of the art.

---

> ### Comment · Reviewer_Yb5B · 2024-11-27
> **Comment**
>
> As authors didn't bother to actually answer my questions, in particular about the actual usage for RS systems, I lower my score.

---

> > ### Author Response · Authors · 2024-11-27
> > **Sorry for the delay of submitting the responses**
> >
> > We are sorry for the delay of submitting the response. Because of the paper revision deadline (later today); we are tying to get one more revision in before we submitting and completing our responses to all revieweres.

---

> ### Author Response · Authors · 2024-11-28
> **Point-by-point response to the comments by Reviewer Yb5B**
>
> We sincerely apologize for the delayed response, as we have tried to submit a few rounds of revisions to address the suggestions and concerns from you and other reviewers before the revision submission deadline.
>
> **Question 1** Since the initial error appears in a paper published at AAAI, why not contact the Program Committee from that edition? I believe AAAI should take responsibility for errors made during their review process.
>
> **Answer:** Thanks for your suggestion. As Reviewer nWHo has pointed out, simply claiming it as being an error is a bit unfair to the original paper as their analysis is a bit informal. The overall result/bound is still correct, but the convergence analysis is missing some necessary/sufficient conditions for the distribution $\theta$. In the revision, we have further clarified this lack of "additional" conditions for the convergence analysis. We have made this point clear in the introduction (Blue text in revision).
>
> Note that one of our main contributions is to provide the first PAC-Bayes bound for the multivariate linear regressions which in special cases reduces to the AAAI paper bound; we also present a sufficient condition by specifying the distributions of $\theta$ (corresponding to $W$ in our bound) under which the bound converges or diverges.  We have presented this result and generalization in new Section 3.
>
> **Question 2:** I'm curious about how similar or different this is to the analysis of random projections and LinUCB. While we're working in a Bayesian framework, the mathematical tools used are quite similar, and some research has already bridged the gap between these approaches (such as the work in https://www.jmlr.org/papers/volume17/14-087/14-087.pdf)
>
> **Answer:** Thanks for providing the reference for the very interesting paper by Russo & Van Roy on information theoretical bound of Thompson sampling. Based on our understanding, both the PAC-Bayes bound and Russo & Van Roy's bound are derived using tools from information theory.
>
> The PAC-Bayes bound, use Alquier's bound [Alquier16] for example, is
>
> \begin{equation*}
>     P\left(\mathbb{E}\_{\theta \sim \rho}[R^{\text{true}}(\theta) - R^{\text{emp}}(\theta)] <  \frac{1}{\lambda}\left[D(\rho||\pi) + \ln \frac{1}{\delta} + \Psi\_{\pi, \mathcal{D}, l}(\lambda, m)\right]\right) \ge 1 - \delta
> \end{equation*}
>
> where $R^{\text{emp}}(\theta)$ is the empirical risk, $R^{\text{true}}(\theta)$ is the true risk, $\pi$ is the prior distribution of $\theta$, $\rho$ is the posterior distribution of $\theta$.
>
> The regret bound of Thompson sampling (Proposition 4, [Russo16]) is
> \begin{equation*}
> \mathbb{E}[R(Y\_{A^\*}) - R(Y_A)] \le \sqrt{\frac{1}{2}I(A^\*, (A; Y))}
> \end{equation*}
> where $A^\*$ is a random variable that maximizes the reward $R$, while $A$ is any other random variable.
>
> We notice some differences between the two bounds:
> 1. In the PAC-bayes bound, $R^{\text{true}}(\theta)$ is the expectation of the random variable $R^{\text{emp}}(\theta)$. Thus the distance between $R^{\text{true}}(\theta)$ and $R^{\text{emp}}(\theta)$ can be measured by concentration inequalities, on which the PAC-bayes bound is based. In the regret bound, $R(Y\_{A^\*})$ is the upper bound of the random variable $R(Y_A)$. The regret bound is deterministic (it is fixed and does not rely on probability), while the PAC-Bayes bound is probabilistic.
>
> 2. The regret bound uses the Pinsker's inequality as its information bound (Step (b) in the proof of Proposition 4, relating to Fact 9  [Russo16]).
> \begin{equation*}
> \mathbb{E}\_{x\sim P}[g(x)] - \mathbb{E}\_{x\sim Q}[g(x)] \le \sqrt{\frac{1}{2}D(P||Q)}
> \end{equation*}
> while the PAC-Bayes bound uses Donsker and Varadhan’s variation (Lemma 2.2, [Alquier21]):
> \begin{equation*}
> \mathbb{E}\_{x\sim P}[g(x)] - \log\mathbb{E}\_{x\sim Q}[e^{g(x)}] \le D(P||Q)
> \end{equation*}
>
> Could you please provide more details regarding the analysis of random projection or LinUCB? We have reviewed the original papers on random projection [Bingham01] and LinUCB [Li10], but we did not find much (mathematical) analysis in them, which has left us uncertain. Please let us know if we have referenced the wrong papers. Thank you for your understanding.
> \
> &nbsp;
>
> References:
>
> [Alquier16] Pierre Alquier, James Ridgway, and Nicolas Chopin. On the properties of variational approximations of gibbs posteriors. Journal of Machine Learning Research, 17(236):1–41, 2016
>
> [Alquier21] Pierre Alquier. User-friendly introduction to pac-bayes bounds. arXiv preprint arXiv:2110.11216, 2021.
>
> [Li10] Lihong Li, et al. A contextual-bandit approach to personalized news article recommendation. Proceedings of the 19th international conference on World wide web. 2010.
>
> [Russo16] Daniel Russo, and Benjamin Van Roy. An information-theoretic analysis of thompson sampling. Journal of Machine Learning Research, 2016.
>
> [Bingham01] Ella Bingham, and Heikki Mannila. Random projection in dimensionality reduction: applications to image and text data. SIGKDD. 2001.

---

> ### Author Response · Authors · 2024-11-28
> **Point-by-point response to the comments by Reviewer Yb5B (Continued)**
>
> **Question 3:** I wonder if your bound could be helpful for optimistic exploration strategies when the user (row) is sampled from a uniform distribution and the item is selected by the algorithm. This might be challenging since independence is broken, and you'd likely need additional assumptions (such as rank-1 matrix only) - which, while commonly used in state-of-the-art approaches, may be overly simplistic.
>
> **Answer:** Thanks for sharing this idea with us! It will be interesting to investigate how to incorporate feedback/responses and item (response) dependency into our bound, which can be non-trivial. We found that one latest paper [Tasdighi24] seems to have an interesting solution on this, where the authors introduce a reinforcement learning algorithm that leverages PAC-Bayes analysis to model epistemic uncertainty in the critic component. This approach enables the algorithm to dynamically adapt its exploration strategy, thereby enhancing performance in continuous control tasks.
>
> **Question 4:** In all EASE-based system implementations I'm aware of, the zero-diagonal constraint is used. While the paper studies this to some extent, can your analysis provide new insights beyond confirming it as a useful bias?
>
> **Answer:** Our bound applies to $W$ with or without zero-diagonal constraint. One effect of the zero-diagonal constraint is that it can reduce the PAC-Bayes bound, thereby improving the model's generalizability.
>
> In details, we have shown in Section 4.3 that if the constraint of $W$ is changed from $\text{diag}(W) \sim N(0, \sigma^2I)$ to zero diagonal constraint $\text{diag}(W) = 0$, then the covariance matrix of each column $\Sigma_r^{1/2}(W^* - W)_{*i}$ will be reduced from $\sigma^2\Sigma\_{r}$ to $\sigma^2(\Sigma_r - (\Sigma\_r^{1/2})\_{*i}(\Sigma_r^{1/2})\_{*i}^T)$, i.e., $\sigma^2\Sigma\_{r} \succeq \sigma^2(\Sigma\_r - (\Sigma_r^{1/2})\_{*i}(\Sigma\_r^{1/2})\_{*i}^T)$. The reduction of the covariance matrix of the distribution of $W$ leads to a smaller PAC-Bayes bound, as shown in the proof of Theorem 4 in Appendix A. This suggests that the model has better generalizability.
>
> **Question 5:** You mention producing 'non-vacuous' bounds. Could you elaborate on what constitutes such a bound? Specifically, if your definition implies practical usefulness, could you explain how these bounds could be used to improve recommender systems?
>
> A PAC-Bayes bound is considered non-vacuous if the gap between the bound and the test error is small. To the best of our knowledge, there is no universal criterion for how small the gap must be to classify a bound as non-vacuous. [Dziugaite17] showed in their experiments that PAC-Bayes bounds within $10$ times the test error can be considered non-vacuous, which is the criterion we adopted.
>
> If a theoretical bound is too vacuous, for example, $1000$ times the test error, then it will predict almost nothing in practice. A discussion of opinions on vacuous bounds can be found in Section 3.1 of [Alquier21].
>
> A non-vacuous bound benefits a recommender system by explaining experimental results. For example, if experiments show that a model has a large test error, the bound can help determine whether this is due to overfitting or a biased test set. However, it is important to note that the bound does not guide model training; it simply justify whether the model, once trained by whatever training method, should perform well on unseen data or not. We have also added a discussion on the empirical implication and potential application of our bounds in Section 6 (Conclusion and Discussions).
>
>
> References:
>
> [Alquier21] Pierre Alquier. User-friendly introduction to pac-bayes bounds. arXiv preprint arXiv:2110.11216, 2021.
>
> [Tasdighi24] Probabilistic Actor-Critic: Learning to Explore with PAC-Bayes Uncertainty, Bahareh Tasdighi, Nicklas Werge, Yi-Shan Wu, Melih Kandemir, arXiv preprint arXiv:2402.03055, 2024.

---

> ### Author Response · Authors · 2024-11-28
> **Point-by-point response to the comments by Reviewer Yb5B (Continued)**
>
> **Weakness 1:** The paper's positioning relative to previous work in recommender systems is inadequate and imprecise. For instance, citing Rendle 2022 for Matrix Factorization/ALS is an unusual choice, as Rendle is primarily known for his 2010 work on Factorization Machines.
>
> **Answer:** Thanks for pointing this out. Indeed, Rendle's 2022 work is less well-known and is a recent effort to revitalize ALS. Based on your suggestion, we have removed it; and we add a more complete references of matrix factorization and matrix completion in the revised Related Works (Appendix D).
>
> **Weakness 2:** The practical applicability of these findings to real-world recommender systems is unclear. While the experimental section successfully demonstrates the theoretical bounds, it falls short of the standards expected in recommender systems research, lacking comprehensive evaluation on metrics and scenarios that matter in practice.
>
> **Answer:** We apologize for the confusion. In Section 6 (Conclusion and Discussions), we have added a detailed discussion on the empirical implications and potential applications of our bounds in recommendation systems. We acknowledge that PAC-Bayesian bounds are derived for surrogate losses (e.g., sum of squared errors), but they can be indirectly related to recommendation evaluation metrics by decomposing the bound into two components: (1) a generalization bound on the surrogate loss, and (2) the empirical correlation between the surrogate loss and top-$k$ metrics.
>
> This connection implies that PAC-Bayes bounds provide guarantees on the generalization of surrogate losses, which is a necessary condition for achieving strong downstream performance. While surrogate losses may not directly align with top-$k$ metrics such as Recall@$k$ or NDCG@$k$, demonstrating low generalization error on the surrogate loss establishes a strong theoretical foundation for the model’s ability to perform well on these evaluation metrics.
>
> For instance, when evaluating a model on finite test sets using metrics like test error, NDCG@$k$, or Recall@$k$, these metrics often fluctuate depending on the specific data in the test set. If a model performs poorly on these metrics, PAC-Bayes bounds can help identify whether the poor results are likely due to model uncertainty (indicated by a large bound) or other factors;  for example: 1) A large PAC-Bayes bound could indicate high model uncertainty, suggesting insufficient training or data sparsity. 2) A small PAC-Bayes bound coupled with poor performance might point to issues like sub-optimal surrogate metrics, data distribution shifts, or model design.
>
> Thus, PAC-Bayes bound can help diagnose and improve model performance more effectively.

---

> ### Author Response · Authors · 2024-11-28
> **Point-by-point response to the comments by Reviewer Yb5B (Continued)**
>
> **Weakness 3:** The analysis primarily focuses on the non-regularized model, which has limited practical relevance as real-world systems invariably use regularization.
>
> **Answers**: Actually, the bound primarily focuses on performance evaluation and does not directly guide the training process. See our reply https://openreview.net/forum?id=XYG98d5bCI&noteId=D4nDlcgnZB. The model $W$ used in the bound **can come from any training process** (whether optimized via EASE, EDLAE, or even obtained from a random initialization. Note that a randomly initialized $W$ is still a LAE model, though its parameters are unoptimized), **with or without regularization**. As long as $W$ is an $n \times n$ matrix, it can be considered as a LAE model, and its performance can be evaluated by the PAC-Bayes bound by comparing the test error and expected test error (true risk).
>
> The relationship between the regularization hyperparameter $\gamma$ and the bound is indirect: Different values of $\gamma$ results in different $W$ (The relationship between $\gamma$ and $W$ can be complicated), and $W$ is a variable in the bound; but $\gamma$ itself is not a variable in the bound. Thus, the bound does not provide guidance on how to choose $\gamma$ to optimize it.
>
> Broadly speaking, PAC-Bayes bound is based on the framework of statistical learning theory [Vapnik99]. In this theory, the regularizer $\gamma\|\|W\|\|_F^2$, with some constant $\gamma$, is equivalent to the constraint $\|\|W\|\|_F^2 < c$ for some $c$, which is part of the structural risk minimization principle [Vapnik91]. The structural risk minimization principle states that as $\gamma$ increases, $c$ decreases. A smaller $c$ leads to a smaller parameter space of $W$ (Note $\|\|W\|\|_F^2 < c$ implies $\|\|W - 0\|\|_F < \sqrt{c}$, meaning the parameter space is a ball of radius $\sqrt{c}$ under the Frobenius norm). Suppose the optimal $W^*$ (the one that minimizes the bound) lies in somewhere of the space. If $c$ is too large, the space of $W$ will be too large, and searching for the optimal $W^*$ will be hard (use any optimization algorithm). If $c$ is too small, the space of $W$ may exclude $W^*$, so you are not able to find it no matter how you search in the space.
>
> However, there is no theoretical guideline for choosing $c$ in statistical learning theory (including in the PAC-Bayes bound). Therefore, $c$ is typically determined empirically, with the general recommendation that it should not be too large or too small.
> \
> &nbsp;
>
> References:
>
> [Vapnik91] Vladimir Vapnik. Principles of risk minimization for learning theory. Advances in neural information processing systems, 4, 1991.
>
> [Vapnik99] Vladimir, Vapnik. The Nature of Statistical Learning Theory. Springer, 1999.

---

> > ### Author Response · Authors · 2024-12-02
> > **Point-by-point response to the comments by Reviewer Yb5B (Continued)**
> >
> > **Question 6:** I suspect the relative tightness shown in the final table is due to M, which bounds the error on the test set. While getting a concentration-like inequality is expected, what additional insights does your analysis provide? From a practical perspective, M significantly reduces the bound. Could you discuss how M is computed (is it updated after each sample or calculated once using all data, including test data)?  Also, a graph showing how the bound and test error vary with the number of observations would be more informative than a table that doesn't indicate how these results compare to the state of the art.
> >
> > **Answer:** Thanks for your suggestions. The computation of $M$ is related to Theorem 7, which we have removed.
> >
> > Due to Reviewer nWHo's suggestion and we also agreed, there is some unfairness in directly comparing Srebro's bound with the new bound as the settings and goals of these bounds are different: the former applies to low-rank weight matrices in element-wise prediction (explicitly recommendation setting), while our bound applies to nearly full-rank weight matrices in vector-wise prediction (implicit recommendation setting). Given this, we have removed Theorem 7. Thus, the only result our model compared with are the training error and testing error, which our bound does not exceed 3x of the test errors in MovieLens 20M, Netflix and MSD and 4x of the test errors in Yelp2018. Note that we have varied the only hyperparameter $\gamma$   to include three additional settings $50, 200, 400$ for validating the robustness of our results. Since we used the real datasets, our sample size $n$ (number of observations) is fixed, we do not include a figure similar to the one in [Germain16], which only uses the synthetic dataset with vary $n$ to compare different bounds, where we use real datasets for evaluation. In addition, we also rechecked the papers [Dziugaite17, Steck19, Steck20] we follow closely in the paper, they also use the table format to present the main results for performance/bound results.
> >
> > &nbsp;
> >
> > References:
> > [Germain16]  Pascal Germain, Francis Bach, Alexandre Lacoste, Simon Lacoste-Julien, PAC-Bayesian Theory Meets Bayesian Inference, NeurIPS 2016
> >
> > [Dziugaite17] Gintare Karolina Dziugaite and Daniel M Roy. Computing nonvacuous generalization bounds for deep (stochastic) neural networks with many more parameters than training data. arXiv preprint arXiv:1703.11008, 2017.
> >
> > [Steck19] Harald Steck. Embarrassingly shallow autoencoders for sparse data. In The World Wide Web Conference, pp. 3251–3257, 2019.
> >
> > [Steck20] Harald Steck. Autoencoders that don’t overfit towards the identity. Advances in Neural Information Processing Systems, 33:19598–19608, 2020.

---

> ### Comment · Reviewer_Yb5B · 2024-12-02
> **Clarifications**
>
> Thanks for the clarifications, I appreciate the updates to the paper
> I still feel than "tight" would be a better wording than "non-vacuous" since I would reserve the later one to scenarios where the prediction made by the bound is actually used to improve a system at a lower cost than cross validation. But if the wording is accepted in the PAC community, I have no strong opposition.  I update my score accordingly.

---

> ### Author Response · Authors · 2024-12-03
> **Thanks for updating the score**
>
> Dear Reviewer Yb5B,
>
> We appreciate you updating the score and apologize for our late response and any inconvenience it may have caused.
>
> Once again, we sincerely appreciate the time and effort you dedicated for reviewing our paper.
>
> Best Regards,
>
> Authors

---

### Official Review · Reviewer_nWHo · 2024-11-04

**Soundness:** 3
**Presentation:** 3
**Contribution:** 3
**Rating:** 6
**Confidence:** 4

**Summary:**

===========================================Post Rebuttal comments=====================

There were very serious problems/errors in the original version of the paper. The authors have worked extremely hard on the rebuttal with spectacular results. The paper has been substantially rewritten and improved based on my comments during the rebuttal period, making **huge progress**.
Therefore, I believe they have achieved the unlikely feat of pushing the paper above the borderline. However, I cannot fully vouch for correctness as I couldn't check all the details of the new proofs, though the direction makes sense and what is written in the rebuttal makes sense.


Summary of original issues and changes/resolutions:


**Problem 1** (solved)

Severe error in Theorem 7 (trivial and worse than existing results due to inconsistent understanding of sampling procedure)


**Solution 1**

Removed Theorem 7


**Problem 2** (More or less solved)

The sampling regime and learning setting didn't make any sense because of the lack of a train test split within each user. This means the results originally only made sense for multivariate regression (though it wasn't proved for that case) and meaningless for RecSys.


**Solution 2**

The authors have modified the whole learning setting and corrected all the theorems and reran the experiments. They have also proved the results for multivariate regression in general.  Solving this issue was one of the key deciding factors in my decision to raise my score.

Nevertheless, it is worth noting that the bounds do depend on quantities such as $\Sigma_{xy}$, which means that although they can be evaluated with empirical estimation, they are difficult to interpret in terms of sample complexity without data dependent assumptions. However, this may well be a general feature of PAC Bayesian bounds in general. Still, in this particular example, it makes the bounds very qualitatively different from other non Bayesian PAC bounds, so the differences could be further discussed. However, I am still ok with the current resolution at this point.


**Problem 3** (solved)

Bounds don't apply naturally to Recsys because of the use of the square loss, which means the bounds cannot really apply to the implicit feedback setting with any reasonable metric such as Recall despite the fact the authors frequently show illustrations of the bounds from the implicit feedback setting.


**Solution 3**

The authors have added many more details and caveats downgrading the originally over the top claims. **Please keep the caveats there in the final version.**


**Problem 4** (solved)

The original literature review was extremely sparse, citing only one work generalization bounds for RecSys/matrix completion.


**Solution 4**

The authors have added a much more detailed literature review based on my comments.


==========================Changes for the individual scores=====================

Main rating: from 3 to 6


Soundness: from 2 to 3

Contribution: from 2 to 3

Presentation: from 2 to 3


============================================Summary===========================

This paper proves generalization bounds for linear auto encoders based on a famous theorem of Alquier [PB2] which extend/transfer those of [PB1] (which apply to the linear regression setting) to the setting of linear auto encoders such as EASE [LA1].  The authors also fix some errors in the convergence analysis of [PB1]: they show that contrary to the claims in [PB1], even with a Gaussian prior, bounds of the family presented in both [PB1]  and the present paper diverge due to the $\psi$ term in Acquire’s bound involving terms of the form $\mathbb{E}(\exp(X^4))$ for a Gaussian $X$. Theorems 1 proves a bound for the Gaussian case, whilst theorems 3 and 4 show the bound for the case where the observations are bounded. Theorems 2 and 5 show that the bounds for theorems 1 and 4 converge to zero as the number of samples tends to infinity, respectively. Theorem 4 is much more general than theorem 1 as it applies to an arbitrary sampling distribution with the property that the observations are bounded. Later in Theorem 6, the authors show how to calculate the bounds from Theorems 1 or 3 more precisely by computing the optimal posterior to minimize the bound, with similar arguments to the calculation of the analytic solution to the optimization problem in [EASE]. Section 4.4 goes further by also calculating the KL divergence between the prior and posterior explicitly, and shows how to estimate the whole bound in practice: unfortunately, the $\Phi$ term in both theorems 1 and 3 is hard to compute in practice, and the authors use a trick from [PB3] to create a coarser upper bound which may not converge as $m\rightarrow \infty$, but has the more favorable property of being easier to calculate. However, it still involves the population expectation of the norm of the row vector of observations for one user, which the authors estimate empirically with the whole dataset. Experiments on real life datasets show that the bounds are not so far from the true generalization gap, which the authors argue makes the bounds non vacuous whilst existing bounds are vacuous.  Further, Theorem 3 shows another bound based on some standard covering number argument approach as in [AR1].

**Strengths:**

1. The paper is reasonably well-written, especially the introduction (though I have grave concerns about the content).
2. The paper corrects a mistake in the informal convergence analysis in [PB1] with a more rigorous analysis of the convergence.
3. The proofs are long and a nice exercise in dealing with various Gaussians and turning them into quadratic forms. The explicit calculation of the posterior in Theorem 6 is also worthy of interest, and most of the proofs I had time to look at seem correct (except theorem 7).

**Weaknesses:**

Note: at ICLR, revisions of the paper can be submitted. I am quite willing to increase my score if the next revision better puts the results in perspective, or provides some clarification on the points I raised.

Before I delve into each weakness individually, here is a summary of the issues:

1. (Fatal) The bounds are **meaningless in a Recommender Systems** scenario for many inter-related reasons, only some least serious of which the authors admit to:
    1. (Partially admitted) The bounds don’t apply to any reasonable loss function or training scenario from Recommender Systems: they don’t work for implicit feedback with measures such as the recall or precision or AUC, and they don’t work for explicit feedback by withholding a part of the interactions either. The loss function is merely the reconstruction error of a new sample (i.e. a new user, with all of its interactions being fed to the model for evaluation) in terms of the square loss. This says absolutely nothing about generalization performance. Perfect performance is achieved by the identity function. In particular, the bounds without the diagonal constraint are meaningless.  The authors do say in the conclusion that “the problem in recommender systems is more complicated….potential for further research”, but this doesn’t do proper justice to exactly how weak the results in the present submission are.
    2. (Not explicitly admitted) Going further in the direction of the point above the first point above, the bounds do not involve any function class restriction apart from Frobenius norm and by the authors’ own admission  the qualitative behaviour of the bounds doesn’t change much with or without the diagonal constraints: there is no non trivial description of the dependence on the number of items $n$, and the rate of decay of in the number of samples $m$ is as weak as $1/m^{1/n}$, in comparison with a more typical $1/\sqrt{m}$.
2. (Serious) Theorem 7 is **always vacuous**: the authors use a covering number argument based on counting the **number of parameters/dimensions** in a space with $m$ dimensions, where $m$ **is the number of samples**. This is despite the fact that the authors claim this result is superior to the result in [AR1], which cannot be the case.
3. (Serious) The **related works** on generalization bounds for recommender systems only includes [AR1], when there are plenty of seminal works in similar directions to [AR1]. Perhaps the authors are trying to dismiss the matrix completion literature because the loss function in this branch of the literature concerns explicit feedback rather than implicit feedback. However, because of the weaknesses above, it would be absurd to claim that the loss function in the present paper is somehow better suited to the recommendation task. Furthermore, [AR1] is not exception to that, so there is no reason not to include more modern works in that direction.
4. (Serious) The authors claim that the bound in [AR1] is vacuous, which I do not believe. They also claim that their own bound is non-vacuous, which I do not believe is a fair statement either. As explained above, if we do not have diagonal constraints, then taking the function class which contains only the identity function gives a vanishingly small bound which is non vacuous in the same sense as the authors’ bound. If we do have diagonal constraints but no specific assumptions on the data, it is not clear how the bounds presented here can take this into account.
5. (minor) Some of the theorem statements somewhat lack clarity/rigor in their presentation.
6. (arguable) Whilst I completely agree with the authors that the analysis in [PB1] is wrong, using words like “error”, however pertinently, when describing other works is dangerous. Certainly, one should be more careful doing it than what the authors are doing when they state (cf. line 114 page 3) “Here the convergence analysis from [PB1]”. Where in the reference is it? After checking, I can see that the authors mean the argument on page 3 after the main theorem. I understand that this analysis indeed constitutes one of the main claims of the paper. However, as a courtesy to the authors, it might be better to rewrite the statements in such a way that it appears as if this is a minor component of the paper. Indeed, to the best of my understanding, there is **no error in [PB1] which is in a Theorem environment**. Only the (admittedly important) description below the theorem is wrong. That is something to capitalize on when crafting a more tactful correction.






***Details:***

On weaknesses 1.1 and 1.2 : the loss function is $\\|r^{\top} -r^{\top} W\\|\_{F}^2$   where $r\in\mathbb{R}^n$ is a vector of interactions for a new user. The generalization bounds in the present paper state that if one is able to reconstruct the training samples well, then one is also able to successfully recover a test set sample. This is assuming the whole sample is fed to the model, so that statement doesn’t contain any information about recommendation performance. It is not clear whether the authors are trying to claim that their model shows generalization bounds for the implicit feedback prediction task (predict which interactions will happen in the unseen test set) or the explicit feedback prediction task (predict the ratings on unseen (user, item) combinations). In the beginning of section 3.4, the authors mention ratings  typically being in the range of [0,5] (line 248 page 5), which hints at the explicit feedback case, but on line 69 in the introduction they hint at the implicit feedback case. At the end, they solve neither: I can understand that solving the implicit feedback case would be challenging, and that the loss function needs to be user by user in an autoencoder setting, but at the absolute minimum, for the sampling strategy to make any sense as a proxy to the real recommendation task, the authors should **split each test user into two parts item-wise**: one to be fed to the model and one to be used at evaluation. For instance, the test error could be defined as $\|r_{test}-r_{train}W\|\_{F}^2$ where $\[r_{train}\]\_j=1$ if $j$ is interacted by the user AND $j$ is in a predetermined “training” subset of items and $\[r_{train}\]\_j=0$ if either of those conditions is not statisfied (similarly, $r_{test}$ should be defined over the complementary, “test” set of item). It is acceptable for the training set to vary randomly for each user, but they must be distinct. The bounds in the present paper will certainly not mean anything in this more rigorous setting: indeed, providing any theoretical insight into why EASE works is a very challenging task which the authors haven’t really attempted: it requires understanding what function class restriction is implicit in the diagonal constraint.  It would probably be easier to prove bounds for EASE like models which introduce a low rank condition (cf. [ELSA]).

(More minor) Furthermore, the approximation the authors use for the practical bounds vaguely appeals to the law of large numbers as a justification. This means that the quantity evaluated by the authors isn’t a bound which they have proved. Why not incorporate the quantitative argument here? Instead of using the whole dataset to estimate $\mathbb{E}(r^\top r)$, the authors should use only the training set and independently prove that this quantity approaches the true value, propagating the errors through the bounds, resulting in a new and similar result which they can evaluate.

On weakness 2:

Ignoring constants,  if $\epsilon $\leq 1$, the bound can be processed this way:

$$4 \left( \frac{8M}{\epsilon} + 1 \right)^{2m} \exp\left( -\frac{m \epsilon^2}{32 M^2} \right) \gtrsim  4\exp(2\log(1+8M) m- \frac{m\epsilon^2}{32M^2})$$

This doesn't converge to zero when $\epsilon$ is less than the constant $8M\sqrt{\log(1+8M)}$.


(This certainly  has to happen given the vacuous argument on page 23, line 1197, which covers a 2m dimensional ball where m is the number of training samples. )

 I think I understand how the authors got confused: there is indeed a covering number over the rows and columns of the matrix in [AR1], which is acceptable in this case because the individual samples are entries in the matrix rather than entire users, which means that the number of observations can be larger than the number of users.


On weakness 3: As explained above, the present paper doesn’t prove meaningful bounds for either implicit or explicit feedback. It appears that the authors are trying to position themselves as the first to have proved meaningful bounds for implicit feedback, or for LAEs, neither of which is true. Thus, it is not clear what branch of the literature should be included. However, if we accept results on explicit feedback, then the whole matrix completion literature should be mentioned. It is worth noting that the only work cited, [AR1], also concerns matrix completion (in a binary classification context), so even if the authors deliberately didn’t’ include the exact recovery literature [MC1,MC2,MC3,IMC] due to the exact observation requirement (or the literature on side information [IMCAR1,2,3,IMC] due to the slightly different setting), it is unclear why the followup works [AR2,AR3,AR4,AR5,AR6,AR7,MAX1,MAX2] were not included, despite treating similar learning settings as [AR1]. Likewise, the recent branch of the literature on the low-noise setting  ([PR1] with explicit rank restriction and [PR2] with nuclear norm regularizers) provides spectacular results in terms of the simultaneous dependence on the noise and the ground truth rank, albeit in the uniform sampling setting only.

On weakness 4: the bound in [AR1], like those of the follow up works, generally scales like $[m+n]r$ in sample complexity where $r$ is the rank. This means the required number of samples for each user is roughly proportional to the rank, up to some constants and log terms. The constants and log terms in [AR1] are not large at all, I cannot believe the bound is vacuous for rank 2 (which achieves reasonably competitive RMSE already).


On weakness 5 (minor) : some theorems are hard to read due to somewhat vague descriptions of the assumptions. For instance, in Theorem 7, the statement “Suppose there exists  $M > 0$ such that $ \|R_i - R_i W\|_F^2 \in [0, M]$  for any $R_i$”  is vague because it appears to make a statement about the training set when what is required is for the inequality to hold with probability one over the test distribution.
Similarly, the definition of $\beta$ in Theorem 4 should be a maximum over the support of the distribution rather than the distribution itself. Similarly, theorem 3 is really a prelude to Theorem 4 more than anything (perhaps it could be a proposition). Further, in Theorem 4, the notation $\eta_1(R)$ to mean the top eigenvalue of the matrix $R$ is used, but it is only introduced in the proof in the supplementary (line 847). It would certainly not hurt to make a table of notations and simple consequences (for instance, explicitly mentioning somewhere that $Q’{Q’}^\top=\Sigma_W$ would make the proof of Theorem 1 more readable. Adding a citation for the inequality on line 682 would also be good form.



***References***


On Pac Bayes

[PB1] V Shalaeva, AF Esfahani, P Germain, M Petreczky, “Improved PAC-Bayesian bounds for linear regression”. AAAI 2020

[PB2] Pierre Alquier. “User-friendly introduction to PAC-Bayes bounds”, Foundations and Trends in Machine Learning, 2021.

[PB3] P Germain, F Bach, A Lacoste, S Lacoste-Julien, “PAC-Bayesian Theory Meets Bayesian Inference”, NeurIPS 2016.

On Linear auto encoders

[EASE] Harold Steck, “Embarrassingly Shallow Autoencoders for Sparse Data”, WWW 2019.

[ELSA] V Vančura, R Alves, P Kasalický, P Kordík,  “Scalable Linear Shallow Autoencoder for Collaborative Filtering”, RecSys 2022


On exact matrix completion (uniform or near uniform sampling)

[MC1] Emmanuel Candes and Terence Tao, “The Power of Convex Relaxation: Near-Optimal Matrix Completion  “, TIT 2009.

[MC2] Benjamin Recht , “A Simpler Approach to Matrix Completion, JMLR 2011”

[MC3] Yudong Chen, Srinadh Bhojanapalli, Sujay Sanghavi, Rachel Ward, “Coherent Matrix Completion”, ICML 2014

On Matrix Completion with noise (including several rank-proxys) under non uniform distributions

[AR1] Nathan Srebro, Noga Alon, and Tommi Jaakkola. “Generalization error bounds for collaborative prediction with low-rank matrices” NeurIPS 2004

[AR2] Nathan Srebro, Russ R. Salakhutdinov. “Collaborative Filtering in a Non-Uniform World: Learning with the Weighted Trace Norm” NeurIPS 2010

[AR3] Nathan Srebro and Adi Shraibman, “Rank, Trace-Norm and Max-Norm”, COLT 2005

[AR4] Rina Foygel, Ruslan Salakhutdinov, Ohad Shamir, Nathan Srebro, “Learning with the Weighted Trace-norm under Arbitrary Sampling Distributions. NeurIPS 2011

[AR5] Ohad Shamir, Shai Shalev-Shwartz,“Collaborative Filtering with the Trace Norm: Learning, Bounding, and Transducing”, COLT 2011

[AR6]Ohad Shamir, Shai Shalev-Shwartz, “Matrix Completion with the Trace Norm: Learning, Bounding, and Transducing”, JMLR 2014

[AR7] Antoine Ledent and Rodrigo Alves, “Generalization Analysis of Deep Non-linear Matrix Completion”, ICML 2024

[MAX1] Rina Foygel, Nathan Srebro, Ruslan Salakhutdinov, “Matrix reconstruction with the local max norm”, NeurIPS 2012

[MAX2] T. Tony Cai, Wen-Xin Zhou,  “Matrix Completion via Max-Norm Constrained Optimization”, Electronic Journal of Statistics 2016.

On matrix completion with side information

[IMC] Miao Xu, Rong Jin, Zhi-Hua Zhou, “Speedup Matrix Completion with Side Information: Application to Multi-Label Learning “, NeurIPS 2013


On matrix completion with side information and noise

[IMCAR1] Kai-Yang Chiang, Cho-Jui Hsieh, Inderjit S. Dhillon, “Matrix Completion with Noisy Side Information”, NeurIPS 2015

[IMCAR2] Kai-Yang Chiang, Cho-Jui Hsieh, Inderjit S. Dhillon, “Using Side Information to Reliably Learn Low-Rank Matrices from Missing and Corrupted Observations”, JMLR 2018

[IMCAR3] Antoine Ledent, Rodrigo Alves, Yunwen Lei and Marius Kloft, “Fine-grained generalization analysis of inductive matrix completion”, NeurIPS 2021


On nearly exact matrix completion with low noise

[PR1] Yuxin Chen, Yuejie Chi, Jianqing Fan, Cong Ma, “Spectral Methods for Data Science: A Statistical Perspective”

[PR2] Yuxin Chen, Yuejie Chi, Jianqing Fan, Cong Ma, Yuling Yan, “Noisy Matrix Completion: Understanding Statistical Guarantees for Convex Relaxation via Nonconvex Optimization”, SIAM J. Opt 2020









***Typos/grammar*** (Minor, non exhaustive)



332: extra capital letter at “We”

Line 669: the sentence is not finished.

Line 247: “ The values…is bounded”…

662: “since… is of multivariate Gaussian…”

Line 626: “the probability that $I-W$ being…” (that> of)

**Questions:**

1. Could you rewrite the paper as suggested below?
2. (minor) could you explain the argument on the second to last line of equation (24)? Since $W^*$ depends on $R^{emp}$, I don’t think it follows from the equation on lines 1155-1156 at face value (though I don’t significantly doubt the big picture of this particular part of the proof).
3. Why do you only ever look at the case $\lambda=m^{1/n}$ instead of $\lambda=\sqrt{m}$? Does the bound break down in that case? It would be better to express the bounds in terms of sample complexity and study how it depends on $n$ as well.


Actionable items to fix the paper at this round or the next (I may increase my score if all of the points below are performed in a revised version which is uploaded) :

1. Unless strong arguments are presented in the rebuttal, it seems clear to me that this paper doesn’t explain the generalization abilities of LAE or any recommender systems method. There is nothing special about the techniques which applies to the reconstruction loss specifically. It would be better to repeat the analysis in the more general context of **multi-output linear regression**, which is what this paper is really about, and to completely change the narrative of the paper to steer clear of any claim of having “the first bounds for EASE” or anything similar. A casual mention of the potential for applications to LAE and why the presented results do not apply as of yet can be left till the end of the paper. If the authors manage to improve the results to cover a different objective as explained in weakness 1, then the relationship with LAE can be reintroduced.
2. Study or at least mention the dependence on $n$
3. Remove theorem 7 altogether
4. Incorporate the approximation from the experiments into a rigorous bound
5. Rerun the experiments on some multi output linear regression model, and present any remaining experiments on RecSys datasets as merely synthetic datasets where the task is different from the recommendation task.

---

> ### Author Response · Authors · 2024-11-25
> **Point-by-point response to the comments by Reviewer nWHo**
>
> Thank you for your insightful suggestions and questions.
>
> **Question 1.1:** Repeat the analysis in the more general context of multi-output linear regression.
>
> **Answer:** We have applied our analysis from LAE to the general multivariate (multi-output) linear regression model and propose a new bound (Theorem 1). It can be shown that Shalaeva's bound is a special case of our bound. Besides, our bound is based on a more general statistical assumption that the multivariate Gaussian distribution of $x$ can be **dependent** or **degenerate**, while in Shalaeva's bound it is assumed to be i.i.d..
>
> **Weakness 4:** The bounds without the diagonal constraint are meaningless.
>
> **Answer:** We completely integrate the constraint $\text{diag}(W) = 0$ to the bound, see Eq (5). The issue is that, the upper bound of part 2 of Eq (5) is difficult to calculate under the constraint $\text{diag}(W) = 0$ since it costs $O(n^4)$. We address this issue by finding an easy-to-calculate upper bound. This upper bound is established by replacing $\text{diag}(W)$ with i.i.d. zero mean Gaussian random variables, is validated by Theorem 4 (This theorem is newly added!), and costs $O(n^3)$.
>
> **Question 4:** Incorporate the approximation of $\mathbb{E}[r^\top r]$ to the main bound.
>
> **Answer:** We tackle this problem by providing two separate error bounds for estimating $\mathbb{E}[r^\top r]$ (which are not incorporated to the main bound): a theoretical bound based on Hoeffding's inequality and an empirical bound based on central limit theorem, see Appendix C. The theoretical bound is vacuous and thus not applicable in practice, while the empirical one is much tighter.
>
> **Weakness 1.2, Question 1.2, Question 3:** The reason why taking $\lambda = m^{1/n}$ and causing $\lambda$ depending on $n$ is not mentioned. Also, why not taking $\lambda=\sqrt{m}$.
>
> **Answer:** We have replaced $n$ with $d$. $\lambda$ can be independent of $n$. The setting $\lambda = m^{1/n}$ is from Shalaeva's paper and we realize it can cause confusion.
>
> It is true that $\lambda=\sqrt{m}$ (i.e., $d = 2$) guarantees the convergence of $m^{-1/d}\ln \mathbb{E}_{\theta \sim \pi} \exp\left(2m^{2/d - 1} v\_{\theta}^2\right)$ as long as $\mathbb{E}\_{\theta \sim \pi}\exp\left(v\_{\theta}^2\right) < \infty$. We keep $d > 2$ because we want to show that the divergence $\lim\_{m\rightarrow \infty} \ln \mathbb{E}\_{\theta \sim \pi} \exp\left(2m^{2/d - 1} v\_{\theta}^2\right) = \infty$ cannot be addressed by taking any $d > 2$ if $\mathbb{E}\_{\theta \sim \pi}\exp\left(t v\_{\theta}^2\right) = \infty$ for any $t > 0$, see Section 3.2 (4).
>
> **Weakness 2, Question 1.3:** Theorem 7 (Srebro's bound) is always vacuous for predicting high dimensional vectors, since it is originally used for low rank weight matrix on element-wise prediction. It should be removed altogether.
>
> **Answer:** We realize that our bound makes an unfair comparison with Srebro's bound, as our bound assumes a nearly full-rank weight matrix (if there is no constraint $\text{diag}(W) = 0$, the Gaussian random matrix $W$ is of full rank [Feng07]) for vector-wise prediction. We have removed Theorem 7.
>
> **Weakness 6:** It is unsuitable to call Shalaeva's convergence analysis an error.
>
> **Answer:** We have revised our description of Shalaeva's convergence analysis, noting that it is not rigorous due to missing conditions.
> \
> &nbsp;
>
> Reference:
>
> [Feng07] Xinlong Feng and Zhinan Zhang. The rank of a random matrix. Applied mathematics and computation, 185(1):689–694, 2007.

---

> > ### Comment · Reviewer_nWHo · 2024-11-28
> >
> > Dear Authors,
> >
> >
> > Many thanks for the revisions! There is certainly a lot of progress after removing theorem 7, improving the conclusion and related works.
> >
> >
> > However, I feel like there is still a misunderstanding regarding the learning setting and the diagonal constraints.
> >
> > To the best of my understanding, as I said in my review, your training samples consist in entire observed users, with the test samples being unobserved users. Thus, without diagonal constraints, the identity function always has perfect performance. It is an extremely large stretch to interpret this in any way as a generalization bound which applies to recommender systems. One solution would be to **split each user into a train and a test by item**: for both training and test users, the task is to predict the last n/2 items based on the first n/2 items, but you optimize the parameters based on the training set. This is still a multi output linear regression problem, so your theorems probably still apply. If you go in this direction at least the bound makes *some* sense for this kind of autoencoders except for the facts you now explain in the conclusion (no notion of recall, wrong loss function, etc.). If you keep the original strategy then the bound makes no sense at all for recsys autoencoders to the best of my understanding.
> >
> >
> > Have you tried running your experiments with such a split?

---

> ### Author Response · Authors · 2024-11-30
> **A Solution to the LAE settings by Reviewer nWHo**
>
> Dear Reviewer nWHo,
>
> Thanks for reminding us! We have realized that the main task of LAE is not to predict the entire vector, but only the hidden items (the items that not shown in training stage but shown in test/evaluation stage) in a vector. **Our bound needs to be revised to adapt to your settings before the experiments can be conducted**. We have developed a new theoretical bound under your settings as well as its practical method for calculation, as shown below:
>
> **Changes in the Main Settings (Section 4.1):**
>
> We first rewrite the empirical risk function from $\frac{1}{m}\|\|R - RW\|\|_F^2$ to $\frac{1}{m}\|\|R^T - WR^T\|\|_F^2$ by taking the transpose and redefine $W^T$ as $W$, so that it aligns with out settings in Section 3. Assume $R\in \\{0, 1\\}^{m\times n}$ and $W \in \mathbb{R}^{n\times n}$.
>
> Now we generate $X \in \\{0, 1\\}^{n\times m}$ and $Y \in \\{0, 1\\}^{n \times m}$ in this way: For any $i \in \\{1, 2, ..., n\\}$ and $j \in \\{1, 2, ..., m\\}$, if $(R^T)\_{ij} = 0$, we set $X\_{ij} = Y\_{ij} = 0$; if $(R^T)\_{ij} = 1$, we set either $X\_{ij} = 1, Y\_{ij} = 0$ or $X\_{ij} = 0, Y\_{ij} = 1$, **each case with $1/2$ probability**. In this way, we randomly split the $1$s in $(R^T)\_{\*j}$ into two groups $X\_{\*j}$ and $Y\_{\*j}$, and the number of $1$s in each group is roughly equal. Note that **$X\_{\*j}$ corresponds to the $r\_{\text{train}}$ and $Y\_{\*j}$ corresponds to $r\_{\text{test}}$ in your settings**.
>
> Suppose we have a LAE model $W$ (The $W$ can be obtained **through any training method**, and we do not focus on the specific approach used, whether it's solved by EASE, EDLAE, or even a randomly initialized one. The constraint $\text{diag}(W) = 0$ can be applied but is not necessary.), its performance is evaluated by the empirical risk
> \begin{equation*}
>     R^{\text{emp}}(W) = \frac{1}{m}\|\|Y - WX\|\|\_F^2 \quad\quad\quad\quad \text{(R1)}
> \end{equation*}
>
> Assume each pair $(X\_{\*j}, Y\_{\*j})$ is i.i.d. sampled from an $2n$ dimensional distribution $\mathcal{D}$, then we can define the true risk as
> \begin{equation*}
>     R^{\text{true}}(W) = \mathbb{E}\_{(x, y)\sim \mathcal{D}}\left[\|\|y - Wx\|\|\_F^2\right] \quad\quad\quad \text{(R2)}
> \end{equation*}
>
> The new bound is obtained by replacing the old bound Eq (5) with the $R^{\text{true}}(W)$ in Eq (R2) and $R^{\text{emp}}(W)$ in Eq (R1).
>
> **Statistical assumptions for $\mathcal{D}$:**
>
> Just like we use Gaussian assumption for $\mathcal{D}$ in Assumption 1 in our paper, we need additional assumptions for the $\mathcal{D}$ in order to build our statistical model. Note that now the support of $\mathcal{D}$ is $\\{0, 1\\}^{2n}$, obviously not Gaussian. Thus we re-assume $\mathcal{D}$ as follows:
> \
> &nbsp;
>  -  **Assumption 3:** Suppose $\mathcal{D}$ is characterized by 3 finite cross-correlation matrices $\Sigma\_{xx} = \mathbb{E}\_{(x, y) \sim \mathcal{D}}[xx^T], \Sigma\_{xy} = \mathbb{E}\_{(x, y) \sim \mathcal{D}}[xy^T]$ and $\Sigma_{yy} = \mathbb{E}\_{(x, y) \sim \mathcal{D}}[yy^T]$, and $\Sigma\_{xx}$ is positive definite.
> \
> &nbsp;
>
> Under Assumption 3, given any $W$, we can expand $R^{\text{true}}(W)$ in the following way:
>
> \begin{align*}
>     &\quad R^{\text{true}}(W) = \mathbb{E}\left[\|\|y - Wx\|\|\_F^2\right] = \sum_{i=1}^n\mathbb{E}[\|\|y\_i - W_{i\*}x\|\|_F^2] = \sum\_{i=1}^n W\_{i\*}\mathbb{E}[xx^T]W\_{i\*}^T - 2W\_{i\*}\mathbb{E}[y\_ix] + \mathbb{E}[y\_i^2] \\\\
> &= \sum\_{i=1}^n  W\_{i\*}\Sigma\_{xx}W\_{i\*}^T - 2W\_{i\*}(\Sigma\_{xy})\_{\*i} + (\Sigma\_{yy})\_{ii} = \sum\_{i=1}^n (W\_{i\*}\Sigma\_{xx}^{1/2})(W\_{i\*}\Sigma\_{xx}^{1/2})^T - 2(W\_{i\*}\Sigma\_{xx}^{1/2})\Sigma\_{xx}^{-1/2}(\Sigma\_{xy})\_{\*i} + (\Sigma\_{yy})\_{ii} \\\\
> &= \sum\_{i=1}^n (W\_{i\*}\Sigma\_{xx}^{1/2} - (\Sigma\_{xy})\_{\*i}^T\Sigma\_{xx}^{-1/2})(W\_{i\*}\Sigma\_{xx}^{1/2} - (\Sigma\_{xy})\_{\*i}^T\Sigma\_{xx}^{-1/2})^T - (\Sigma\_{xy})\_{\*i}^T\Sigma\_{xx}^{-1}(\Sigma\_{xy})\_{\*i} + (\Sigma\_{yy})\_{ii} \\\\
> &= \sum\_{i=1}^n \|\|W\_{i\*}\Sigma\_{xx}^{1/2} - (\Sigma\_{xy})\_{\*i}^T\Sigma\_{xx}^{-1/2}\|\|\_F^2 - \|\|\Sigma\_{xx}^{-1/2}(\Sigma\_{xy})\_{\*i}\|\|\_F^2 + (\Sigma\_{yy})\_{ii} \\\\
> &= \|\|W\Sigma\_{xx}^{1/2} - \Sigma\_{xy}^T\Sigma\_{xx}^{-1/2}\|\|\_F^2 - \|\|\Sigma\_{xy}^T\Sigma\_{xx}^{-1/2}\|\|_F^2 + \text{tr}(\Sigma\_{yy}) \quad\quad\quad\quad \text{(R3)}
> \end{align*}

---

> ### Author Response · Authors · 2024-11-30
> **A Solution to the LAE settings by Reviewer nWHo (Continued)**
>
> To calculate the bound, we use Assumption 2 from our paper: Let $W$ be a Gaussian random matrix, with its prior distribution $\pi$ being $\bar{\mathcal{N}}(\mathcal{U}_0, \sigma^2J)$ and its posterior distribution $\rho$ being $\bar{\mathcal{N}}(\mathcal{U}, \mathcal{S})$.
>
> **Changes in calculating the optimal $\rho$ (Section 4.2):**
>
> The optimal $\rho$ obtained by Theorem 3 is revised as follows:
>
> Without constraint $\text{diag}(W) = 0$, the optimal $\rho$ is given by
> \begin{equation*}
>     \mathcal{S}\_{ij} = \frac{1}{\frac{2\lambda}{m}X\_{i*}X_{i\*}^T + \frac{1}{\sigma^2}} \text{ for } i, j \in \\{1, 2, ..., n\\}
> \end{equation*}
> \begin{equation*}
>     \mathcal{U} = \left(\frac{1}{m}XX^T + \frac{1}{2\lambda\sigma^2}I\right)^{-1}\left(\frac{1}{m}YX^T + \frac{1}{2\lambda\sigma^2}\mathcal{U}_0\right)
> \end{equation*}
>
> with constraint $\text{diag}(W) = 0$, the optimal $\rho$ is given by
> \begin{equation*}
>     \mathcal{S}_{ij} = \frac{1}{\frac{2\lambda}{m}X\_{i\*}X\_{i*}^T + \frac{1}{\sigma^2}} \text{ for } i, j \in \\{1, 2, ..., n\\}, i\ne j \quad\quad \mathcal{S}\_{ii} = 0 \text{ for } i \in \\{1, 2, ..., n\\}
> \end{equation*}
> \begin{equation*}
>     \mathcal{U} = \left(\frac{1}{m}XX^T + \frac{1}{2\lambda\sigma^2}I\right)^{-1}\left(\frac{1}{m}YX^T + \frac{1}{2\lambda\sigma^2}\mathcal{U}\_0 - \frac{1}{2}\text{Diag}(x)\right)
> \end{equation*}
> where
> \begin{equation*}
>     x = 2 \cdot\textnormal{diag}\left[\left(\frac{1}{m}XX^T + \frac{1}{2\lambda\sigma^2}I\right)^{-1}\left(\frac{1}{m}YX^T + \frac{1}{2\lambda\sigma^2}\mathcal{U}\_0\right)\right] \oslash \textnormal{diag}\left[\left(\frac{1}{m}XX^T + \frac{1}{2\lambda\sigma^2}I\right)^{-1}\right]
> \end{equation*}
>
> The proof is similar to the original one.
>
> **Changes in calculating $\Psi$ (Section 4.3):**
>
> Since by Eq (R3), $R^{\text{true}}(W) = \|\|W\Sigma\_{xx}^{1/2} - \Sigma\_{xy}^T\Sigma_{xx}^{-1/2}\|\|\_F^2 - \|\|\Sigma\_{xy}^T\Sigma\_{xx}^{-1/2}\|\|\_F^2 + \text{tr}(\Sigma\_{yy})$, denote $B = \Sigma\_{xy}^T\Sigma_{xx}^{-1/2}$ and $c = - \|\|\Sigma_{xy}^T\Sigma\_{xx}^{-1/2}\|\|\_F^2 + \text{tr}(\Sigma\_{yy})$, then
> \begin{equation*}
>     R^{\text{true}}(W) = \sum\_{i=1}^n\|\|W\_{i\*}\Sigma\_{xx}^{1/2} - B\_{i\*}\|\|\_F^2 + c \quad\quad\quad\quad \text{(R4)}
> \end{equation*}
>
> If $W \sim \bar{\mathcal{N}}(\mathcal{U}\_0, \sigma^2J)$ with $\text{diag}(W) = 0$ constraint, then $W\_{\*i}^T \sim \mathcal{N}((\mathcal{U}\_0)\_{i\*}^T, \sigma^2(I - I^i))$ and $(W\_{i\*}\Sigma_{xx}^{1/2} - B\_{i\*})^T \sim \mathcal{N}(\Sigma_{xx}^{1/2}(\mathcal{U}\_0)\_{i\*}^T - B\_{i\*}^T , \sigma^2(\Sigma\_{xx} - [(\Sigma\_{xx})\_{i\*}^{1/2}]^T[(\Sigma\_{xx})\_{i\*}^{1/2}]))$.
>
> Therefore, if in Section 4.3 we redefine $\mu^i = \Sigma\_{xx}^{1/2}(\mathcal{U}\_0)\_{i\*}^T - B\_{i\*}^T$ and $A = \sigma^2\Sigma\_{xx}$, then the upper bound of $\Psi$ given by Theorem 4 can be calculated in the same way.
>
> Let $A = \sigma^2\Sigma\_{xx} = S^T\Lambda S$ be the eigenvalue decomposition where $\Lambda = \text{diag}(\eta_1, \eta_2, ..., \eta_n)$. The upper bound of $\Psi$ now becomes
> \begin{align*}
> &\Psi\_{\pi, \mathcal{D}}(\lambda, m) \le \ln \mathbb{E}\left[e^{R^{\text{true}}(W)}\right] = \ln \mathbb{E}\left[\exp\left(\|\|W\Sigma\_{xx}^{1/2} - B\|\|\_F^2 + c\right)\right] = \ln \mathbb{E}\left[\exp\left(\sum\_{i=1}^n\|\|W\_{i\*}\Sigma\_{xx}^{1/2} - B\_{i\*}\|\|\_F^2 + c\right)\right] \\\\
> &= c + \ln \mathbb{E}\left[\exp\left(\sum\_{i=1}^n\|\|W\_{i\*}\Sigma\_{xx}^{1/2} - B\_{i\*}\|\|\_F^2\right)\right] \le c + \ln \prod\_{i=1}^n \frac{\exp\left(\frac{\frac{1}{\sigma^2}\lambda\eta_i\|\|S_{i\*}(\mathcal{U}_0^T - \Sigma\_{xx}^{-1/2}B^T)\|\|_F^2}{1 - 2\lambda \eta\_i}\right)}{(1 - 2\lambda \eta\_i)^{n/2}}
> \end{align*}
>
> If without the constraint $\text{diag}(W) = 0$, the last $\le$ above becomes $=$.

---

> ### Author Response · Authors · 2024-11-30
> **A Solution to the LAE settings by Reviewer nWHo (Continued)**
>
> **Experiments:**
>
> We obtain $W$ through minimizing the EASE function:
> \begin{equation*}
>     \min_{W}\|\|X - WX\|\|_F^2 + \gamma \|\|W\|\|_F^2 \quad\quad \text{s.t. } \text{diag}(W) = 0  \quad\quad\quad\quad \text{(R5)}
> \end{equation*}
> But note that this is not the only method to obtain $W$. We just use it as an example.
>
> We also need to approximate $\Sigma\_{xx}, \Sigma\_{xy}$ and $\Sigma\_{yy}$, which is carried out using the same way in Section 4.4. Suppose there exists a test set $(X', Y')$ with $X' \in \\{0, 1\\}^{n \times m'}$ and $Y' \in \\{0, 1\\}^{n \times m'}$ and $(X\_{\*j}, Y\_{\*j})$ is i.i.d. from $\mathcal{D}$, Denote $X\cup X' = [X; X'] \in \\{0, 1\\}^{n\times (m + m')}$ and $Y\cup Y' = [Y; Y'] \in \\{0, 1\\}^{n\times (m + m')}$, then we have
> \begin{align*}
>     \Sigma\_{xx} &\approx \hat{\Sigma}\_{xx} = \frac{1}{m + m'} [X \cup X'][X \cup X']^T \\\\
>     \Sigma\_{xy} &\approx \hat{\Sigma}\_{xy} = \frac{1}{m + m'} [X \cup X'][Y \cup Y']^T \\\\
>     \Sigma\_{yy} &\approx \hat{\Sigma}\_{yy} = \frac{1}{m + m'} [Y \cup Y'][Y \cup Y']^T
> \end{align*}
>
> The error of the approximation above can be measured with the bounds in Appendix C of our paper. The $B$ and $c$ in Eq (R4) can be calculated with the approximated $\Sigma\_{xx}, \Sigma\_{xy}$ and $\Sigma\_{yy}$.
>
> Denote ${R^{\text{emp}}}'(W) = \frac{1}{m'}\|\|Y' - WX'\|\|_F^2$, i.e., $R^{\text{emp}}(W)$ is the test error on the training set while ${R^{\text{emp}}}'(W)$ is the test error on the test set. **Both $R^{\text{emp}}(W)$ and ${R^{\text{emp}}}'(W)$ are unbiased estimators of $R^{\text{true}}(W)$ so we can apply PAC-Bayes bound on them**. The evaluation of the bound is implemented by **comparing the gap between the PAC-Bayes bound and ${R^{\text{emp}}}'(W)$ (or $R^{\text{emp}}(W)$)**.
>
> **Remarks:**
>
> 1. The Assumption 3 is very general since it not only includes the case when $\mathcal{D}$ is of binary support, like $\\{0, 1\\}^{2n}$, but also any bounded support, like $[a, b]^{2n}$.
>
> 2. Since we use EASE function Eq (R5) to obtain $W$, the training error can be expressed as $\frac{1}{m}\|\|X - XW\|\|_F^2$. However, it is hard to build a bound between the training error $\frac{1}{m}\|\|X - XW\|\|_F^2$ and the test error $\frac{1}{m}\|\|Y - XW\|\|_F^2$ since there are of different metrics and there seems to be no direct statistical connections between them.
>
> 3. It is also unnecessary to build a bound between the training error $\frac{1}{m}\|\|X - XW\|\|_F^2$ and the test error $\frac{1}{m}\|\|Y - XW\|\|_F^2$. As we mentioned in the answer of Question 5 of Reviewer Yb5B, the $W$ can be obtained by whatever training method, and PAC-bayes bound only evaluates the model's test performance. Since the PAC-Bayes bound can be independent of the training process, it is meaningless to build relationship between training error and testing error with a PAC-bayes bound, especially when training error and test error are under different metrics.
>
> \
> &nbsp;
> &nbsp;
>
> Please let us know if this solution is acceptable or you have any suggestions. Also, **please give us one more day to carry out and post experiment results**, as we need to rewrite programs and redo experiments. Thanks again for your patience!

---

> ### Author Response · Authors · 2024-12-01
> **Experimental Results**
>
> Dear Reviewer nWHo,
>
> Thanks for your patience! Below are our experimental results on MovieLens 20M and Netflix datasets.
> \
> &nbsp;
>
> **MovieLens 20M:**
> | $\gamma\quad\quad$ | $R^{\text{emp}}(W)$ | ${R^{\text{emp}}}'(W)$ | PAC-Bayes bound
> | --- | --- | --- | --- |
> | 50 | 66.43 | 59.53 | 122.53 |
> | 100 | 53.24 | 53.71 | 97.27 |
> | 200 | 59.83 | 54.00 | 109.47 |
> | 400 | 56.51 | 51.74 | 102.97 |
>
> &nbsp;
>
> **Netflix:**
> | $\gamma\quad\quad$ | $R^{\text{emp}}(W)$ | ${R^{\text{emp}}}'(W)$ | PAC-Bayes bound
> | --- | --- | --- | --- |
> | 50 | 82.20 | 79.20 | 139.66 |
> | 100 | 84.58 | 80.97 | 143.27 |
> | 200 | 80.82 | 77.45 | 135.73 |
> | 400 | 78.76 | 75.79 | 131.65 |
>
> &nbsp;
>
> $\gamma$ is the hyperparameter of EASE, and we set $\gamma = 50, 100, 200, 400$ to train different EASE models $W$ for evaluation. The experimental results show that our bound is tight, not exceeding $3\times$ of $R^{\text{emp}}(W)$ or ${R^{\text{emp}}}'(W)$.
>
> The settings used in the experiment are mostly the same as in our paper: We split the $85\\%$ samples from the entire dataset into the training set $R$ and the rest $15\\%$ into the test set $R'$. Then we generate $X, Y$ from $R$ and $X', Y'$ from $R'$. The only difference is that we have transformed the range of ratings in the datasets from $[0, 5]$ to $[0, 1]$ by setting all non-zero entries to $1$. This change results in smaller test errors compared to the experimental results in our paper.
>
> We use an $80-20$ splitting strategy ($80\\%$ of the 1s in $R \cup R'$ randomly assigned to $X \cup X'$ and $20\\%$ to $Y \cup Y'$) for both MovieLens 20M and Netflix. It needs to be mentioned that both MovieLens 20M and Netflix are sparse datasets, and MovieLens 20M is sparser than Netflix. We observed an issue with MovieLens 20M dataset: using a $50-50$ splitting strategy makes $X \cup X'$ too sparse, causing $\hat{\Sigma}\_{xx} = \frac{1}{m + m'} [X \cup X'][X \cup X']^T$ to be singular. Since $\Sigma\_{xx}^{-1/2}$ in Eq (R3) is approximated by $\hat{\Sigma}\_{xx}^{-1/2}$, a singular $\hat{\Sigma}\_{xx}$ will result in an undefined $\hat{\Sigma}\_{xx}^{-1/2}$, which makes the bound **uncomputable**. Thus one limitation when calculating the theoretical bound in practice is that **$X \cup X'$ must be sufficiently dense to ensure that $\hat{\Sigma}_{xx}$ is non-singular**.
>
> It should also be noticed that $\hat{\Sigma}\_{xx}^{-1/2}$ being singular does not imply $\Sigma_{xx}^{-1/2}$ is singular. While $\Sigma\_{xx}^{-1/2}$ is unknown to us, if its non-singularity holds true, then given more data, we can obtain a non-singular approximation $\hat{\Sigma}\_{xx}^{-1/2}$ for $\Sigma\_{xx}^{-1/2}$. This suggests that **the singularity of $\hat{\Sigma}\_{xx}$ is likely due to insufficient data for a non-singular approximation**. Our experiments on MovieLens 20M support this. We found that if $X'$ is not included in calculating $\hat{\Sigma}\_{xx}$, i.e., simply use $\hat{\Sigma}\_{xx} = \frac{1}{m}XX^T$, then a $95 - 5$ splitting fails. If including $X'$, then an $80 - 20$ splitting works, while a $70 - 30$ splitting fails. This implies that adding more data can help address the singularity issue in the approximation $\hat{\Sigma}\_{xx}$.
>
> In summary, if the approximation $\hat{\Sigma}\_{xx} = \frac{1}{m}XX^T$ is singular, we suggest two ways for addressing this issue: (1) Split more 1s to $X$, ensuring that $X$ is sufficiently dense to guarantee the non-singularity of $\frac{1}{m}XX^T$. (2) Introduce additional data $X'$ (sampled from the same distribution as $X$) and set $\hat{\Sigma}\_{xx}$ as $\frac{1}{m+ m'}[X \cup X'][X \cup X']^T$.
>
> \
> &nbsp;
>
> If you are satisfied with these updated results, we would be happy to include them in the paper. We kindly ask for your feedback or any further suggestions.

---

> > ### Author Response · Authors · 2024-12-03
> >
> > Dear Reviewer nWHo,
> >
> > We sincerely thank you for your insightful and constructive suggestions. Per your suggestion, we have improved our PAC-Bayes bound to the multivariate linear regression scenario, making the results much more general. Additionally, for the evaluation metrics, we have provided a detailed argument demonstrating how our PAC-Bayes bound can be used to help address uncertainty and performance discrepancies of the recommendation model. Furthermore, based on your suggestion on the training/testing item split setting, we have obtained new analytic and experimental results of the PAC-Bayes bound under such a setting.
> >
> > As the discussion deadline approaches, we kindly ask whether your main concerns have been addressed and if there are any other questions of interest that you would like to raise.
> >
> > We greatly appreciate the time and effort you have devoted to reviewing our work, and your detailed suggestions to help improve the paper!
> >
> > Best Regards,
> >
> > Authors

---

### Official Review · Reviewer_gE7b · 2024-11-06

**Soundness:** 3
**Presentation:** 3
**Contribution:** 3
**Rating:** 6
**Confidence:** 1

**Summary:**

This paper investigates PAC-Bayes bounds for linear autoencoders and presents two distinct bounds. The first bound is developed under the assumption of a Gaussian data distribution, while the second bound is based on bounded data distributions and Gaussian parameter assumptions. To make these theoretical results more accessible for practical applications, the authors introduce a simplified upper bound for the second case. They then adapt this upper bound to the EASE (Efficient and Accurate Sampling-based Estimation) recommender system, demonstrating its effectiveness through experimental validation on standard recommender datasets.

**Strengths:**

- The paper provides a rigorous and detailed analysis of PAC-Bayes bounds, complete with comprehensive proofs (though the correctness of these proofs has not been independently verified). The theoretical framework is well-developed, contributing meaningfully to the field of statistical learning theory.
- By extending the theoretical bounds to the EASE model, the authors bridge the gap between theory and practice, making the results applicable to real-world problems. The evaluation on recommender system datasets demonstrates the practical relevance and potential impact of their approach.
- The paper is well-organized, with clear explanations and a strong motivation for the research. The structure allows readers to follow the logic and understand the significance of the contributions.

**Weaknesses:**

-  Given that there may be space available, consider moving the related work section from the appendix (if present) to the main text. Furthermore, a more in-depth discussion on how the proposed bounds compare to existing work would make the paper more comprehensive and inclusive.
- It is unclear whether the proposed PAC-Bayes bounds can be generalized to all linear autoencoder models. The paper should clarify whether these bounds are specific to the conditions outlined or if they have broader applicability across different linear autoencoder architectures.
- The theoretical results are intriguing, but the paper could benefit from a more detailed discussion of what these PAC-Bayes bounds imply for practical recommender system applications. Specifically, how do these bounds inform model selection, regularization strategies, or error expectations in practice?

**Questions:**

- One of the main questions is whether the practical bounds derived in this work can be applied to all linear autoencoder models. For example, is it possible to calculate a general PAC-Bayes bound for any linear autoencoder using Theorem 6? If there are limitations or constraints, it would be valuable to elaborate on them to clarify the scope of the theoretical results.
- The methodology used to calculate the PAC-Bayes bound in Table 2 requires further elaboration. Did you use a fixed \epsilon in your calculations? If so, what was the reasoning behind this choice, and how might different \epsilon values affect the results? Additionally, how sensitive are the practical training and test errors to hyperparameter choices? If there are significant fluctuations, a discussion on how to interpret these variations (e.g., how fluctuations influence conclusion about 2x or 4x test error) would be very helpful.
- Given that the practical performance of recommender systems is often influenced by hyperparameter settings, how should readers interpret the PAC-Bayes bounds in light of these variations? If hyperparameter tuning introduces substantial variability, how do these fluctuations affect the practical utility of the derived bounds?

---

> ### Author Response · Authors · 2024-11-28
> **Point-by-point response to the comments by Reviewer gE7b**
>
> We would like to first thank you for your insightful reviews. Our responses to your questions and concerns are listed below.
>
> **Question 1, Weakness 2:** It is unclear whether the proposed PAC-Bayes bounds can be generalized to all linear autoencoder models. The paper should clarify whether these bounds are specific to the conditions outlined or if they have broader applicability across different linear autoencoder architectures. For example, is it possible to calculate a general PAC-Bayes bound for any linear autoencoder using Theorem 6? If there are limitations or constraints, it would be valuable to elaborate on them to clarify the scope of the theoretical results.
>
> **Answer:** Our PAC-bayes bound can be generalized to any LAE model $W$, including those with the zero diagonal constraint $\text{diag}(W) = 0$, as discussed in our reply https://openreview.net/forum?id=XYG98d5bCI&noteId=D4nDlcgnZB . In fact, any $n \times n$ matrix $W$, regardless of how it is obtained, is an LAE model, and our bound can be applied to it. Several prominent LAE-based recommender system models like EASE, EDLAE and ELSA adopt zero diagonal constraint, to which our bound is also applicable.
>
> One limitation of our bound is that some LAE models may impose specific constraints on $W$, making the bound difficult to compute. For example, some LAE models require $W$ to be of low rank. While the bound can still be formed for a low rank $W$, its calculation can be difficult (We may end up with a theoretical bound that cannot be computed). This is because, the calculation of the bound relies Assumption 2 in our paper, where we assume $W$ to be a random Gaussian matrix ($W$ does not have to be Gaussian, but assuming it to be Gaussian makes the bound easy to calculate, while other distributions may not). A random Gaussian matrix $W$ is of full rank [Feng07] and cannot be reduced to the low rank case.
>
> Another limitation is that our bound uses square loss (Frobenius norm) as the metric for the error between target and prediction, since the square loss is easy for statistical analysis. Many real-world recommender systems, however, use other metrics like NDCG@K, Recall@K, and our bound cannot be directly be applied to these metrics. Moreover, these metrics are challenging to analyze from a statistical perspective.
>
> **Question 2.1:** The methodology used to calculate the PAC-Bayes bound in Table 2 requires further elaboration. Did you use a fixed $\epsilon$ in your calculations? If so, what was the reasoning behind this choice, and how might different $\epsilon$ values affect the results?
>
> **Answer:** Thanks for your suggestions. The calculation of $\epsilon$ is related to Theorem 7, which we have removed. Due to Reviewer nWHo's suggestion and we also agreed, there is some unfairness in directly comparing Srebro's bound  with the new bound as the settings and goals of these bounds are  different: the former applies to low-rank weight matrices in element-wise prediction (explicitly recommendation setting), while our bound applies to nearly full-rank weight matrices in vector-wise prediction (implicit recommendation setting). Given this, we have removed Theorem 7.
>
> **Question 2.2:** How sensitive are the practical training and test errors to hyperparameter choices? If there are significant fluctuations, a discussion on how to interpret these variations (e.g., how fluctuations influence conclusion about 2x or 4x test error) would be very helpful.
>
> **Answer:** We added experiments where we set $\gamma$ (the only hyperparameter of EASE, see Eq (2)) to 50, 200, and 400. The results show that our bound remains within 3 times the test error on MovieLens 20M, Netflix, and MSD, and 4 times the test error on Yelp2018, despite fluctuations. Thus, this demonstrates our results are not sensitive due to the hyperparameter choices.
>
> (Update) We have some new experimental results on a different LAE setting, which can be found here: https://openreview.net/forum?id=XYG98d5bCI&noteId=T7jUjy5Ige
> \
> &nbsp;
>
> Reference:
>
> [Feng07] Xinlong Feng and Zhinan Zhang. The rank of a random matrix. Applied mathematics and computation, 185(1):689–694, 2007.

---

> ### Author Response · Authors · 2024-11-28
> **Point-by-point response to the comments by Reviewer gE7b (Continued)**
>
> **Question 3, Weakness 3:** How the bound explains the fluctuation in performance caused by hyperparameter tuning, and how it informs model selection, regularization strategies, or error expectations.
>
> **Answer:** The bound does not directly guide the training process of a model; instead, it provides information about the model's test performance. Specifically, once we obtain a model $W$ from any training method (whether optimized via EASE, EDLAE, or even obtained from a random initialization. Note that a randomly initialized $W$ is still a LAE model, though its parameters are unoptimized.), as long as $W$ an $n\times n$ matrix, it can be treated as a LAE model, and we can evaluate its test performance with the bound.
>
> The changes of hyperparameters affect the bound indirectly: Different hyperparameter results in different $W$, and $W$ is a variable of the bound. There is no direct relationship between the hyperparameters and the bound, as the hyperparameters are not included as variables in the bound.
>
> The bound does not provide guidance on hyperparameter tuning or regularization strategies, as these relate to the training process. The bound informs model selection in a way that a model $W$ with a smaller PAC-bayes bound is better, since it has better generalizability, i.e., likely to make more accurate prediction on unseen data. Regarding error expectations, the bound provides a probabilistic relationship between test error and the expected test error (true risk); it is independent of training error.
>
> **Weakness 1.1:** Given that there may be space available, consider moving the related work section from the appendix (if present) to the main text.
>
> Thanks for your suggestions. We have significantly revised the paper and there is no additional space for related works due to the 10-page limit. We have also expanded the related works by adding papers related to the matrix completion and trace norm.
>
> **Weakness 1.2:** A more in-depth discussion on how the proposed bounds compare to existing work would make the paper more comprehensive and inclusive.
>
> To the best of our knowledge, the PAC-Bayes bounds for linear regression are primarily covered in two papers. [Germain16] proposed the first bound for multiple linear regression (single dependent variable), but the bound does not guarantee convergence. [Shalaeva20] improved upon [Germain16]'s bound by ensuring convergence, although the conditions required for convergence are not discussed. Both [Germain16] and [Shalaeva20] focus on single-output linear regression. In the revision, we have proposed the first PAC-Bayes bound for multivariate linear regression, where the bound in [Shalaeva20] is a special case of our general bound. In addition, we have provided sufficient conditions that guarantee convergence of both our bound and Shalaeva's bound. Finally, we have applied our bound for study LAE on the zero-diagonal case. We have highlighted and clarified the above problems and contributions/differences in the revised Section 1 (Introduction).
> \
> &nbsp;
>
> References:
>
> [Germain16] Pascal Germain, Francis Bach, Alexandre Lacoste, and Simon Lacoste-Julien. Pac-bayesian theory
> meets bayesian inference. Advances in Neural Information Processing Systems, 29, 2016.
>
> [Shalaeva20] Vera Shalaeva, Alireza Fakhrizadeh Esfahani, Pascal Germain, and Mihaly Petreczky. Improved
> PAC-bayesian bounds for linear regression. In Proceedings of the AAAI Conference on Artificial
> Intelligence, volume 34, pp. 5660–5667, 2020.

---

> > ### Author Response · Authors · 2024-12-03
> >
> > Dear Reviewer gE7b,
> >
> > We appreciate the diligent efforts of you in evaluating our paper. We have carefully addressed your concerns on the limitations and practical benefits of the PAC-Bayes bound and have provided additional experimental results per your suggestion.
> >
> > As the discussion period is coming to a close, please let us know if our responses have resolved all of your questions.
> >
> > Thank you once again for your time and consideration.
> >
> > Best Regards,
> >
> > Authors

---

### Author Response · Authors · 2024-11-25
**Revised Paper: PAC-Bayes Bounds for Multivariate Linear Regression and Linear Autoencoders**

Dear Reviewers,

We sincerely appreciate the insightful suggestions and constructive feedback you have provided in your individual reviews. Based on your input, we have made significant revisions to the paper, with all changes highlighted in blue for clarity. Specifically, we have addressed the following major points:

**1**. (Critical) Section 3: We have written a new section on PAC-Bayes bounds for multivariate linear regression. We have generalized our theoretical PAC-Bayes bound from LAE to multivariate (multi-output) linear regression, as presented in Theorem 1. We show that Shalaeva's bound for single-output linear regression is a special case of our bound. Additionally, our bound is based on a more general statistical assumption that the multivariate Gaussian distribution of $x$ can be **dependent** or **degenerate**, whereas Shalaeva's bound assumes it to be i.i.d. In Section 3.2, we have also updated the convergence analysis to accommodate the new bound.

**2**. (Important) Section 4: In the method of calculating the bound for EASE, we have modified our previous assumptions on $\text{diag}(W)$ from zero mean Gaussian random variables to zero constants, thus completely incorporating the constraint $\text{diag}(W) = 0$ to the bound. In Section 4.2, Theorem 3 is adjusted to reflect this change.

- (Critical) In Section 4.3, we highlight that the constraint $\text{diag}(W) = 0$ makes the $\Psi$ term difficult to calculate due to $O(n^4)$ complexity. To address this issue, we propose a new theorem (Theorem 4) showing that an upper bound of $\Psi$ can be obtained by replacing $\text{diag}(W)$ with zero mean Gaussian random variables, and this upper bound costs only $O(n^3)$, making it more practical to compute. The proof of Theorem 4 is particularly interesting, as we find the Loewner partial order [Horn12] very helpful in constructing it.

- In Section 4.4, we have fixed a typo in the formula of $\text{approx}(\Psi')$: the $1/2$ is modified to $n/2$. This modification increases the bound slightly, and we have re-done all the experiments in Section 5.

**3**. Section 1: We have clarified the main challenges and specific problems for PAC-Bayes bound of LAEs, including the need for multivariate linear regression, the need to add additional condition for convergence proof and the need to address the calculation difficulties introduced by the zero diagonal constraint of LAE models used in practice. We have also revised the main contribution.

**4**. Section 2: we have added a description of multivariate linear regression and have detailed the relationship between the regularizer and constrained optimization.

**5**. (Important) Section 6: We have modified the conclusion and add further discussions on 1) applying PAC-Bayes bound for other LAE modes; 2) Empirical implications and potential applications of PAC-Bayes bound for recommendation settings; 3) an important open problem in connecting recommendation model (surrogate) loss and top $k$ evaluation metrics systematically and analytically.

**6**. (Important) Appendix A: We have revised the proofs of Theorem 1 and Theorem 2 to improve clarity. Specifically, in the proof of Theorem 2, we replaced the source of the Dominated Convergence Theorem from Resnick's book [Resnick98] with Rudin's book [Rudin76], making the proof easier to understand.

- (Important) Appendix C: We have added a theoretical bound based on Hoeffding's Inequality and an empirical bound based on central limit theorem to describe the approximation error of $\mathbb{E}_{r\sim \mathcal{D}}[rr^T]$ in Section 4.3, and show that the theoretical bound is much looser than the empirical bound, making it difficult to apply in practice.

- Appendix D: We have added some related works on matrix factorization and matrix completion.

**7**. (Important) Section 5: We have added experimental results for $\gamma$ (the hyperparameter of EASE) set to 50, 200, and 400. Additionally, we have included a clarification on the criterion for determining the non-vacuousness of a bound. The revised code has been uploaded to the supplementary material.

Please let us know if you have any further comments and questions. Thanks!
\
&nbsp;

References:

[Horn12] Roger A Horn and Charles R Johnson. Matrix Analysis. Cambridge university press, 2012.

[Rudin76] Walter Rudin. Principles of mathematical analysis. McGraw-hill New York, 1976.

[Resnick98] Sidney I Resnick. A probability path. Birkhauser Boston, 1998.

---

### Meta-Review · Area_Chair_PUfb · 2024-12-16

**Metareview:**

The paper provides a PAC-Bayesian analysis of linear autoencoder models, motivated by their success in recommender system settings. This involves extending existing analyses of multiple linear regression to multivariate linear regression, and incorporating additional constraints common in the recommender setting into the bounds.

The initial reviews raised several concerns in the initial submission, notably around the viability of the original Theorem 7, the precise applicability of the results to the recommender system setting (as opposed to generic multivariate linear regression), and several questions regarding technical details (such as the precise incorporation of diagonal constraints).

Following the response, the authors uploaded a revision with several changes, notably removing the original Theorem 7, and rewriting some key sections (Section 3.1, Section 4.1, Section 4.3, bound calculation in Section 5). These changes are to the benefit of the paper, and appear to significantly address many of the reviewers' concerns. Indeed, the reviewers were satisfied with the new version, with the final recommendations being weakly supportive of acceptance.

At the same time, the revision is arguably substantially different from the original submission (akin to a Major rather than Minor revision). Indeed, there are significant changes to every major section, and even the main claim of the paper has seen a change (with greater emphasis on multivariate regression, and linear autoencoders being a corollary). Given this, we are hesitant to unreservedly recommend publication -- we would prefer if the paper undergoes a fresh round of review, where the claims can be carefully examined.

_Minor comment_: In addition to EASE, it is recommended to cite the SLIM work of Lin and Karypis, "Sparse Linear Methods for Top-N Recommender Systems", ICDM 2011.

**Additional Comments On Reviewer Discussion:**

Per above, the discussion period saw significant changes to the paper, which indeed positively addressed several of the reviewer concerns. Reviewers were unanimously in (weak) support of the paper following this. However, given the magnitude of the changes, we are hesitant to unreservedly recommend publication.

---

### Decision · Program_Chairs · 2025-01-22

Reject